# Deep Flow Networks

**Ozan Candogan** [1]    **Ayoub Foussoul** [1]

## Abstract

We introduce Deep Flow Networks (DFNs), a new class of discrete function approximators. DFNs are inspired by and generalize minimum-cost flow value functions that map node imbalances on a subset of nodes to the optimal flow cost. Such functions are known to be M-convex (Murota, 2003) and admit efficient optimization. On the theoretical side, we prove that DFNs are universal approximators for discrete functions on $\mathbb{Z}^d$ that admit convex extensions to $\mathbb{R}^d$, and characterize their optimization complexity in terms of their deviation from the M-convex regime. Guided by these results, we develop a practical DFN implementation for learning from data. Finally, we evaluate our implementation empirically on data from different ground-truth functions, showing that DFNs achieve strong approximation accuracy while being substantially faster to optimize than benchmark approaches.

## 1. Introduction

### 1.1. Motivation and Background

A common paradigm in machine learning is *predict-then-optimize*. In this setting, we first learn a model that maps inputs $x$ to outcomes $y$. We then use the model within a downstream optimization problem to make decisions. This pipeline is common for two fundamental reasons.

First, the ground-truth mapping $x \mapsto f(x)$ may be *computationally expensive* to evaluate, often because it is given by the solution value of a complex optimization problem. In such cases, one learns a surrogate that approximates $f$ and then optimizes the surrogate instead. For example, (Kody et al., 2022) replace the nonlinear AC power-flow constraints in the unit-commitment problem with a

learned neural-network surrogate and solve the resulting formulation as a mixed-integer linear program (MILP). Similarly, (Kaleem et al., 2025) replace the vehicle-routing cost term in a location–routing model with a learned neural-network surrogate and solve the resulting formulation as a MILP. In inventory control, (Harsha et al., 2025) learn a neural approximation of the cost-to-go and compute replenishment decisions by solving a MILP. However, these approaches often yield MILP reformulations that are large and only weakly structured. As a result, they can still be difficult to solve in practice.

Second, the underlying mapping may be *unknown* altogether (or only partially observed), and only data are available. In revenue management, for instance, one observes customer purchase decisions rather than latent valuations or an explicit demand function. A standard approach is to fit a parametric choice model (e.g., Logit) and then perform pricing or assortment optimization over the learned surrogate (Talluri & Van Ryzin, 2004; Train, 2009; Besbes & Zeevi, 2009). While these classical models often lead to tractable downstream optimization, they can impose rigid structural assumptions that limit approximation power.

These considerations motivate the need for models that are simultaneously good approximators and tractable for downstream optimization. For continuous decision variables, one representative approach is to use universal convex function approximators that admit efficient continuous optimization; for example, (Calafiore et al., 2019) develop log-sum-exp approximators that can approximate any convex function on compact set and that are easy to optimize over continuous domains. Closely related is max-affine regression (Ghosh et al., 2022), for which log-sum-exp serves as a smooth approximation.

However, in many applications—including the power generation, vehicle routing, and inventory problems mentioned above—the decision variables are inherently discrete (e.g., generator on/off decisions, vehicle routing decisions, or integer replenishment quantities). Crucially, convexity over $\mathbb{R}^d$ does not translate into tractability over $\mathbb{Z}^d$: minimizing a convex function over integer points is NP-hard in general, even for convex quadratic objectives over $\{0,1\}^d$ (Gritzmann & Klee, 1989). To the best of our knowledge, existing predict-then-optimize approaches over integer decisions

---

[1]Booth School of Business, University of Chicago, Chicago, IL, USA. Correspondence to: Ayoub Foussoul <ayoub.foussoul@chicagobooth.edu>.

*Proceedings of the 43$^{rd}$ International Conference on Machine Learning*, Seoul, South Korea. PMLR 306, 2026. Copyright 2026 by the author(s).

typically rely on generic surrogate families, rather than surrogates specifically designed for downstream integer optimization.

The focus of this paper is on approximating discrete ground-truth functions that are convex-extendible while allowing efficient downstream optimization. A discrete function $f : \mathbb{Z}^d \to \mathbb{R} \cup \{+\infty\}$ is convex-extendible if there exists a convex function $g : \mathbb{R}^d \to \mathbb{R} \cup \{+\infty\}$ such that $g(x) = f(x)$ for all $x \in \mathbb{Z}^d$. This class covers many discrete functions that arise in practice, including value functions from dynamic programs in inventory management (often $L^\natural$-convex) (Chen, 2017), network-flow value functions (Murota, 2003), and customer valuation functions with the gross substitutes property in discrete-choice modeling (Kelso & Crawford, 1982). In particular, we propose a new class of function approximators designed to both accurately fit discrete convex-extendible objectives and enable efficient downstream optimization over integer domains.

### 1.2. Additional Related Literature

One related strand of literature is on function approximators for set-valued or binary inputs, such as Deep Sets (Zaheer et al., 2017), Deep Submodular Functions (Dolhansky & Bilmes, 2016), and Extended Deep Submodular Functions (Hosseini et al., 2024). While these are discrete-function approximators, they target set-valued or binary domains rather than general integer domains. These are also not designed to facilitate post-training optimization, and lead to complex generic downstream optimization formulations.

Another related strand of literature is on surrogate-optimization models, especially Gaussian-process and Kriging models (Jones et al., 1998), as well as radial-basis-function models (Gutmann, 2001). These models are, however, more commonly used in sequential loops that alternate model fitting, optimization, and new data acquisition than in the fixed-surrogate predict-then-optimize setting considered here.

Our paper is also related to the literature on discrete convex optimization. Convex extendability alone is generally too weak to yield interesting optimization guarantees, since minimizing a convex-extendible discrete function over integer points is NP-hard in general, even for convex quadratic objectives over $\{0, 1\}^d$ (Gritzmann & Klee, 1989). Accordingly, several more structured subclasses have been studied, including submodular set functions (Lovász, 1983), $L/L^\natural$-convex and $M/M^\natural$-convex functions (Murota, 2003), and integrally convex functions (Favati & Tardella, 1990). Minimum-cost-flow value functions form a particularly important subfamily of $M$-convex functions (Murota, 2003), combining algorithmic tractability with sufficient richness to approximate complex data, hence our choice of these

subfamilies as the building block of DFN.

### 1.3. Contributions

We introduce *Deep Flow Networks* (DFNs), a class of function approximators that is inspired by and generalizes minimum-cost flow value functions that map node imbalances on a subset of nodes to the optimal flow cost. Specifically, the node imbalances in a DFN are given by an affine function $Ax + b$ of the input $x$, where $A$ is an integer matrix and $b$ is an integer vector. When $A = I$, the resulting functions are $M$-convex and can be efficiently minimized. DFNs extend these functions by allowing a general integer matrix $A$. We show the following properties for DFNs.

**Universality.** We show that DFNs can approximate (up to an affine scaling) any convex-extendible discrete function on a finite domain, covering a broad family of important functions both in theory and practice including set submodular, $M/L$-convex, and integrally convex functions.

**Optimization of trained DFNs.** We characterize the computational complexity of minimizing a trained DFN in terms of a parameter $\Delta(A)$ that measures the deviation of the learned matrix $A$ from the "easy" $M$-convex regime ($A = I$). We prove that DFNs satisfy a *generalized exchange property* (Theorem 2.3), extending the classical $M$-convex exchange property, and use it to bound the optimization complexity as a function of $\Delta(A)$ (Lemma 2.4). Hence, by controlling $\Delta(A)$, one can retain tractability of downstream integer optimization of the trained DFN functions.

**Practical implementation and empirical validation.** We design a training procedure for DFNs that differentiates through the minimum-cost flow layer and learns integer-valued parameters via straight-through estimation (STE) (Section 3). To bias learning toward the $M$-convex tractable regime, we initialize the matrix $A$ "near" the identity. We evaluate DFNs on supervised regression tasks generated by three different convex-extendible discrete functions, assessing both predictive accuracy and the speed/quality of downstream integer optimization. Our results show that DFNs achieve competitive predictive performance while being substantially faster to optimize over integer domains than benchmark ReLU multi-layer perceptron (MLP) and LogSumExp models.

### 1.4. Notation

Given a function $f : \mathbb{Z}^d \to \mathbb{R} \cup \{\pm\infty\}$, its effective domain is $\operatorname{dom} f := \{ x \in \mathbb{Z}^d : -\infty < f(x) < +\infty \}$.

For a matrix $A \in \mathbb{Z}^{p \times d}$, let $\operatorname{rank}(A)$ denote the rank of $A$. If $A$ is square ($p = d$), let $\det(A)$ denote its determinant. In general, let $\det^*(A)$ denote the pseudo-determinant of

$A$, defined as the product of the nonzero singular values of $A$. For $r \leq \min\{p, d\}$, let $\mathrm{diag}_{p,d}(s_1, \ldots, s_r)$ denote the $p \times d$ matrix with $(i, i)$-entry equal to $s_i$ for $1 \leq i \leq r$ and all other entries equal to $0$.

For a vector $x \in \mathbb{R}^d$, define the coordinate-wise positive and negative parts by $x^+ := \max\{0, x\}$ and $x^- := \max\{0, -x\}$, where the maxima are taken component-wise. We write $\|x\|_1 := \sum_i |x_i|$, $\|x\|_\infty := \max_i |x_i|$, and $\|x\|_2 := \left(\sum_i x_i^2\right)^{1/2}$ for the $\ell_1$, $\ell_\infty$, and Euclidean norms of $x$, respectively. For indices $1 \leq i \leq j \leq n$, let $x_{i:j} := (x_i, \ldots, x_j)^\top \in \mathbb{R}^{j-i+1}$ denote the subvector of $x$ formed by its coordinates $i, \ldots, j$.

## 2. Deep Flow Networks

### 2.1. Architecture Definition

A *Deep Flow Network (DFN)* is a tuple $\Pi = (G, c, u, S, A, b)$, where $G = (V, E)$ is a directed graph with $n := |V|$ nodes and $m := |E|$ directed edges; $c \in \mathbb{Q}^m$ is the vector of edge costs; $u \in \mathbb{Z}_{\geq 0}^m$ is the vector of edge capacities; $S = \{v_1, \ldots, v_p\} \subseteq V$ is a set of $p$ special nodes; and $A \in \mathbb{Z}^{p \times d}$ and $b \in \mathbb{Z}^p$ are an integer matrix and vector, respectively.

Given a DFN $\Pi = (G, c, u, S, A, b)$ and an integer input vector $x \in \mathbb{Z}^d$, let $f(x; \Pi)$ denote the optimal value of the minimum-cost flow problem on $G$ with costs $c$, capacities $u$, and where each special node $v_i \in S$ has a net outflow $a_i^\top x + b_i$ (where $a_i^\top$ is the $i$-th row of $A$ and $b_i$ is the $i$-th entry of $b$), and every other node $v \in V \setminus S$ has zero net outflow. Equivalently, $f(x; \Pi)$ is the optimal value of the integer program

$$\min_\xi \quad c^\top \xi$$

$$\text{s.t.} \quad \sum_{e \in \delta_{v_i}^+} \xi_e - \sum_{e \in \delta_{v_i}^-} \xi_e = a_i^\top x + b_i, \quad \forall i \in [p], \quad (1a)$$

$$\sum_{e \in \delta_v^+} \xi_e - \sum_{e \in \delta_v^-} \xi_e = 0, \quad \forall v \in V \setminus S, \quad (1b)$$

$$0 \leq \xi_e \leq u_e, \ \xi_e \in \mathbb{Z}, \quad \forall e \in E, \quad (1c)$$

where $\delta_v^+ := \{(v, w) \in E\}$ and $\delta_v^- := \{(w, v) \in E\}$. We adopt the convention that $f(x; \Pi) = +\infty$ if (1) is infeasible. We refer to $f(\cdot; \Pi) : \mathbb{Z}^d \to \mathbb{R} \cup \{+\infty\}$ as the *DFN function* induced by $\Pi$. We remark that (1) can be solved in polynomial time using standard minimum-cost flow algorithms (Ahuja et al., 1993).

### 2.2. Universality

In this section, we show that DFN functions can approximate (up to an affine scaling) any convex function on a finite integer domain. In particular, we have the following theorem:

**Theorem 2.1.** *Given a convex function* $g : \mathbb{R}^d \to \mathbb{R} \cup \{+\infty\}$, *a finite set of integers* $I \subset \mathrm{dom}\, g \cap \mathbb{Z}^d$ *and* $\epsilon > 0$, *there exists a DFN* $\Pi$ *and constants* $\alpha, \beta \in \mathbb{R}$ *such that*

$$\max_{x \in S} |g(x) - (\alpha f(x; \Pi) + \beta)| \leq \epsilon$$

*for every* $x \in I$.

The proof of Theorem 2.1 is given in Appendix A.1. Theorem 2.1 implies that DFN functions can approximate (up to an affine scaling) any convex-extendible discrete function on a finite set. This class includes many important functions that arise in practice, including submodular set functions (Lovász, 1983), $M$- and $M^\natural$-convex functions (Murota, 2003), $L$- and $L^\natural$-convex functions (Murota, 2003), and integrally convex functions (Favati & Tardella, 1990).

### 2.3. Optimization of DFN Functions

Consider a DFN $\Pi = (G, c, u, A, b)$. The goal of this section is to characterize the complexity of minimizing the induced DFN function $f(\cdot; \Pi)$ as a function of its parameters. In particular, we define a parameter $\Delta(A)$ which depends on the matrix $A$ and show that the complexity of minimizing $f(\cdot; \Pi)$ scales with $\Delta(A)$.

When $A = I$, the induced DFN function $f(\cdot; \Pi)$ is known to be $M$-convex. In particular, it satisfies the following exchange property:

**Lemma 2.2** (EXC). *For every* $x, x' \in \mathrm{dom}\, f(\cdot; \Pi)$ *and every* $i \in [p]$ *with* $x_i > x'_i$, *there exists* $j \in [p]$ *with* $x_j < x'_j$ *such that*

$$f(x; \Pi) + f(x'; \Pi) \geq f(x - e_i + e_j; \Pi) + f(x' + e_i - e_j; \Pi)$$

In words, whenever $x$ has a unit surplus over $x'$ in some coordinate $i$, there exists a coordinate $j$ in which $x$ has a unit deficit relative to $x'$ such that performing the elementary swap $\tilde{x} = e_i - e_j$ on $x$ (and the opposite swap on $x'$)—thereby bringing the two points closer—keeps both points feasible and does not increase the sum of their objective values. This can be viewed as a discrete analogue of classical convexity where moving two points toward their "average" does not increase the sum of their values. It is also analogous to the classical exchange axiom of matroids; see (Oxley, 2011). A proof of Lemma 2.2 can be found in (Murota, 2003); see also Lemma 2.5 where we prove a stronger exchange property for this class of functions. As a consequence, $f(\cdot; \Pi)$ can be minimized efficiently. In particular, (Shioura, 2022) shows that the steepest descent algorithm starting at any feasible solution and iteratively selecting the best-improving neighbor among points $x + e_i - e_j$, $i, j \in [p]$ reaches a global minimum of $f(\cdot; \Pi)$ in polynomial time.

When $A$ is a general integer matrix however, $f(\cdot; \Pi)$ need not be $M$-convex, and the steepest-descent algorithm of

(Shioura, 2022) is no longer guaranteed to find a global minimizer of $f(\cdot; \Pi)$. We show below that $f(\cdot; \Pi)$ satisfies a generalized exchange property involving exchanges between $\Delta(A)$ coordinates, where $\Delta(A) = 2$ when $A = I$ (recovering the exchange property in Lemma 2.2) and $\Delta(A)$ grows as $A$ moves farther from the $A = I$ regime. Based on this property, we establish a local descent algorithm similar to that of (Shioura, 2022) in which each iteration moves to an improving point in a neighborhood whose size depends on $\Delta(A)$, then link its time complexity to the maginitude of the parameter $\Delta(A)$.

Let $k = p - \text{rank}(A)$ denote the (row) corank of $A$ and let

$$\Delta(A) = 2 \cdot (2k)^{2k} \cdot \det^*(A)$$

where $\Delta(A) = 2 \det^*(A)$ when $k = 0$, i.e., when $A$ is full row rank. The DFN function $f(x; \Pi)$ has the following Generalized Exchange property (G-EXC).

**Theorem 2.3** (G-EXC). *For every $x, x' \in \text{dom } f(\cdot; \Pi)$ such that $Ax \neq Ax'$, there exists $\tilde{x} \in \mathbb{Z}^d$ such that,*

$$f(x; \Pi) + f(x'; \Pi) \geq f(x - \tilde{x}; \Pi) + f(x' + \tilde{x}; \Pi).$$

*Moreover, $0 < \|A\tilde{x}\|_1 \leq \Delta(A)$, $(A\tilde{x})^+ \leq (Ax - Ax')^+$, and $(A\tilde{x})^- \leq (Ax - Ax')^-$.*

The proof of Theorem 2.3 is given in the next subsection (Section 2.4). Given the generalized exchange property (G-EXC), the global minimum of $f(\cdot; \Pi)$ can be found using the local descent Algorithm 1.

---

**Algorithm 1** Local Descent

---

   **Initialize:** $x \in \text{dom } f(\cdot; \Pi)$.
   **while** true **do**
      $\tilde{x} \leftarrow 0$
      **for** $y \in \mathbb{Z}^p$ s.t. $\|y\|_1 \leq \Delta(A)$ **do**
         **if** $\exists \hat{x} \in \mathbb{Z}^d$ s.t. $A\hat{x} = y$ **and** $f(x + \hat{x}; \Pi) < f(x; \Pi)$
         **then**
            $\tilde{x} \leftarrow \hat{x}$; **break**
         **end if**
      **end for**
      **if** $\tilde{x} = 0$ **then**
         **return** $x$
      **end if**
      $x \leftarrow x + \tilde{x}$
   **end while**

---

In particular, the following lemma holds:

**Lemma 2.4.** *Algorithm 1 finds a global minimizer of $f(\cdot; \Pi)$ and terminates in time polynomial in $p^{\Delta(A)}$, $n$, $d$, $D$, $\|c\|_\infty$, $\|u\|_\infty$, $\log \|A\|_\infty$, and $\log \|b\|_\infty$, where $D$ is the smallest positive integer such that $Dc_e \in \mathbb{Z}$ for every $e \in E$.*

The proof of Lemma 2.4 is given in Appendix A.2. Lemma 2.4 shows that the complexity of minimizing $f(\cdot; \Pi)$ is closely related to the corank of $A$ and its pseudo-determinant. This relationship can be leveraged in supervised regression with DFN functions to learn approximators that are easy to optimize over integer domains. In our implementation, we leverage the relationship by initializing the matrix $A$ close to the identity matrix (Section 3).

### 2.4. Proof of Theorem 2.3

Let $\Pi^* = (G, c, u, I, b)$ denote the DFN obtained from $\Pi$ by replacing $A$ with the identity matrix $I$. In particular, $f(x; \Pi) = f(Ax; \Pi^*)$ for every $x \in \mathbb{Z}^d$. As mentioned earlier, $f(\cdot; \Pi^*)$ is $M$-convex and therefore satisfies the exchange property in Lemma 2.2.

We next show that $f(\cdot; \Pi^*)$ in fact satisfies an even stronger exchange property, which we call the *strong base-orderable exchange (SBO-EXC)* property. This is an analogue of *strong base-orderability* for matroids, where a matroid is said strongly base-orderable if, for any two bases $B$ and $B'$, there exists a bijection $\psi : B \to B'$ such that for every subset $X \subseteq B$, both $(B \setminus X) \cup \psi(X)$ and $(B' \setminus \psi(X)) \cup X$ are bases (Oxley, 2011).

**Lemma 2.5** (SBO-EXC). *For any distinct $y, y' \in \text{dom } f(\cdot; \Pi^*)$, there exist indices $i_1, \ldots, i_K$ and $j_1, \ldots, j_K$ in $[p]$ such that*

$$(y - y')^+ = \sum_{t=1}^{K} e_{i_t} \qquad and \qquad (y - y')^- = \sum_{t=1}^{K} e_{j_t}.$$

*Letting $\tilde{y}_t := e_{i_t} - e_{j_t}$ for $t = 1, \ldots, K$, and $\tilde{y}_T := \sum_{t \in T} \tilde{y}_t$ for every $T \subseteq [K]$, it also holds that,*

$$f(y; \Pi^*) + f(y'; \Pi^*) \geq f(y - \tilde{y}_T; \Pi^*) + f(y' + \tilde{y}_T; \Pi^*).$$

In words, Lemma 2.5 says that the unit surpluses of $y$ over $y'$ can be paired with the unit deficits of $y$ relative to $y'$ to obtain $K$ elementary swaps $\tilde{y}_t = e_{i_t} - e_{j_t}$, such that applying any subset of the swaps to $y$ (and the opposite swaps to $y'$) does not increase the sum of the objective values of the two points. The proof of Lemma 2.5 is given in Appendix A.3.

Now, let $x, x' \in \text{dom } f(\cdot; \Pi)$ such that $Ax \neq Ax'$, and set $y := Ax$ and $y' := Ax'$. Let $\tilde{y}_1, \ldots, \tilde{y}_K$ denote the elementary swaps given by Lemma 2.5 applied to $y, y' \in \text{dom } f(\cdot; \Pi^*)$. Assume there exists $|T| \leq \Delta(A)/2$ and $\tilde{x} \in \mathbb{Z}^d$ such that $A\tilde{x} = \tilde{y}_T$, then

$$A(x - \tilde{x}) = y - \tilde{y}_T, \qquad A(x' + \tilde{x}) = y' + \tilde{y}_T,$$

and by Lemma 2.5,

$$f(x; \Pi) + f(x'; \Pi) \geq f(x - \tilde{x}; \Pi) + f(x' + \tilde{x}; \Pi).$$

Moreover, since each swap satisfies $\|\tilde{y}_t\|_1 = 2$, we have $\|A\tilde{x}\|_1 = \|\tilde{y}_T\|_1 \leq 2|T| \leq \Delta(A)$. Also, $(A\tilde{x})^+ = (\tilde{y}_T)^+ \leq (y - y')^+$ and $(A\tilde{x})^- = (\tilde{y}_T)^- \leq (y - y')^-$, implying the theorem. The goal henceforth is therefore to find a subset $T$ of cardinality $|T| \leq \Delta(A)/2$ such that the system of Diophantine equations $A\tilde{x} = \tilde{y}_T$ has a solution over integers $\tilde{x} \in \mathbb{Z}^d$.

The standard method to solve a system of Diophantine equations uses the Smith normal form (SNF) of $A$.

**Definition 2.6** (Smith Normal Form). Let $A \in \mathbb{Z}^{p \times d}$ with $r = \text{rank}(A)$. Then $A$ admits a factorization $A = VDU$, where $V \in \mathbb{Z}^{p \times p}$ and $U \in \mathbb{Z}^{d \times d}$ are unimodular integer matrices (i.e. $\det(U) = \pm 1$ and $\det(V) = \pm 1$), and

$$D = \text{diag}_{p,d}(s_1, \ldots, s_r) \in \mathbb{Z}^{p \times d}, \quad s_1, \ldots, s_r \geq 1.$$

Given an SNF $A = VDU$, the following claim, whose proof is immediate, characterizes when the system has an integer solution.

*Claim* 2.7. The system $A\tilde{x} = \tilde{y}_T$ has a solution over integers $\tilde{x} \in \mathbb{Z}^d$ iff

$$(V^{-1}\tilde{y}_T)_i \text{ is a multiple of } s_i \quad \forall i \in \{1, \ldots, r\}.$$

and

$$(V^{-1}\tilde{y}_T)_{r+1:p} = 0.$$

Now, since the system has a solution for $T = [K]$ (as $\tilde{y}_{[K]} = y - y' = A(x - x')$), we have by Claim 2.7 that

$$\sum_{t=1}^{K} (V^{-1}\tilde{y}_t)_{r+1:p} = (V^{-1}\tilde{y}_{[K]})_{r+1:p} = 0.$$

We now apply the following lemma, also known as Steinitz' lemma, to reorder the vectors

$$(V^{-1}\tilde{y}_1)_{r+1:p}, \ldots, (V^{-1}\tilde{y}_K)_{r+1:p},$$

whose sum is zero, in a way that all the resulting prefix sums have a norm bounded by a quantity that is independent of the the number of vectors $K$, implying that for a large enough $K$, two prefix sums must be equal and hence their difference gives a subset of these vectors whose sum is zero.

**Lemma 2.8** (see, e.g., (Bárány, 2008)). *Let $(\mathbb{R}^k, \|\cdot\|)$ be any $k$-dimensional normed space with unit ball $B := \{z \in \mathbb{R}^k : \|z\| \leq 1\}$. If $v_1, \ldots, v_K \in B$ and $\sum_{i=1}^{K} v_i = 0$, then there exists a permutation $\pi$ of $\{1, \ldots, K\}$ such that for every $t = 1, \ldots, K$,*

$$\left\| \sum_{i=1}^{t} v_{\pi(i)} \right\| \leq k.$$

Let $M_i := \max_{j=1,\ldots,p} |(V^{-1})_{ij}| \geq 1$ for every $i = r + 1, \ldots, p$, and let

$$T := \text{diag}\left( \frac{1}{2M_{r+1}}, \ldots, \frac{1}{2M_p} \right).$$

Note that for every $t = 1, \ldots, K$ and $i = r + 1, \ldots, p$,

$$\left| (V^{-1}\tilde{y}_t)_i \right| = \left| (V^{-1}e_{i_t} - V^{-1}e_{j_t})_i \right| \leq 2M_i.$$

Hence, for every $t = 1, \ldots, K$,

$$\|T(V^{-1}\tilde{y}_t)_{r+1:p}\|_\infty \leq 1.$$

Applying Steinitz' lemma to the vectors $v_t := T(V^{-1}\tilde{y}_t)_{r+1:p} \in \mathbb{R}^k$ in the normed space $(\mathbb{R}^k, \|\cdot\|_\infty)$, there exists a permutation $\pi$ of $[K]$ such that for every $t = 1, \ldots, K$,

$$\left\| \sum_{s=1}^{t} v_{\pi(s)} \right\|_\infty \leq k.$$

Equivalently, for every $t = 1, \ldots, K$ and $i = r+1, \ldots, p$,

$$-2kM_i \leq \sum_{s=1}^{t} (V^{-1}\tilde{y}_{\pi(s)})_i \leq 2kM_i.$$

Define the prefix sums $\delta_t := \sum_{s=1}^{t} V^{-1}\tilde{y}_{\pi(s)} \in \mathbb{Z}^p$ for $t = 1, \ldots, K$. We view $\delta_t$ in the product group

$$\left( \mathbb{Z}/s_1\mathbb{Z} \right) \times \cdots \times \left( \mathbb{Z}/s_r\mathbb{Z} \right) \times \mathbb{Z}^k.$$

i.e., the first $r$ coordinates of $\delta_t$ are taken modulo $(s_1, \ldots, s_r)$ and the last $k$ coordinates are kept in $\mathbb{Z}^k$. Since $\mathbb{Z}/s_1\mathbb{Z} \times \cdots \times \mathbb{Z}/s_r\mathbb{Z}$ has exactly $\prod_{i=1}^{r} s_i$ elements, and since the last $k$ coordinates of $\delta_t$ lie in a box of side lengths $4kM_i$, the number of possible values of $\delta_t$ in the product group is at most

$$N := \left( \prod_{i=1}^{r} s_i \right) \cdot (4k)^k \cdot \prod_{i=r+1}^{p} M_i.$$

Therefore, either $K \leq N$, in which case we take $T = [K]$, or else there exist indices $0 \leq \ell < t \leq K$ such that $\delta_\ell = \delta_t$ in the product group. In the latter case,

$$\sum_{s=\ell+1}^{t} V^{-1}\tilde{y}_{\pi(s)} = 0$$

$$\text{in } \left( \mathbb{Z}/s_1\mathbb{Z} \right) \times \cdots \times \left( \mathbb{Z}/s_r\mathbb{Z} \right) \times \mathbb{Z}^k.$$

By Claim 2.7, this implies that $\sum_{s=\ell+1}^{t} \tilde{y}_{\pi(s)}$ lies in the integer image of $A$, i.e., there exists $\tilde{x} \in \mathbb{Z}^n$ such that

$$A\tilde{x} = \sum_{s=\ell+1}^{t} \tilde{y}_{\pi(s)}.$$

and we pick $T := \{\pi(\ell + 1), \ldots, \pi(t)\}$. In both cases, $|T| \leq N$ and we have a solvable system $A\tilde{x} = \tilde{y}_T$.

We conclude by showing the following lemma.

**Lemma 2.9.** *The Smith normal form $A = VDU$ can be chosen such that*

$$\prod_{i=r+1}^{p} M_i = \prod_{i=r+1}^{p} \max_{j=1,\ldots,p} |(V^{-1})_{ij}| \leq k^k \cdot \frac{\det^*(A)}{\prod_{i=1}^{r} s_i}.$$

The proof of the lemma is given in Appendix A.4.

## 3. Implementation

We now describe our implementation for training DFN functions on a supervised learning task.

### 3.1. Fixed Hyperparameters

**Layered topology.** In principle, one could take $G$ to be a complete directed graph and let learning effectively delete edges by driving capacities to zero. In our experience, this is computationally expensive and not very stable. Instead, we use a layered directed architecture.

Concretely, we choose a number of layers $H \geq 2$ with $\ell_1, \ldots, \ell_H$ nodes each. Let $L_h$ denote the set of nodes in layer $h$, with $|L_h| = \ell_h$. Between consecutive layers $L_h$ and $L_{h+1}$ we include a complete directed bipartite subgraph, i.e., all arcs $(v, w)$ and $(w, v)$ for $v \in L_h$ and $w \in L_{h+1}$. We take the special node set to be $S := L_1 \cup L_H$, so that input-dependent net outflows are injected/removed only at the boundary layers, while intermediate layers enforce flow routing.

**Guaranteeing flow balance.** To ensure that the minimum-cost flow instance defining $f(\cdot; \Pi)$ is feasible for every input $x$, the total net outflow must sum to zero. We enforce this by adding one additional *balancing* node to the final layer $L_H$ whose net outflow is $-\sum_{s_i \in S} (a_i^\top x + b_i)$, thereby canceling the total net outflow of the special nodes.

**Guaranteeing routability.** Even when total net outflow is zero, infeasibility can occur if capacities prevent routing. To avoid this, we add a complete bipartite set of arcs between $L_1$ and $L_H$ (arcs in both directions). These arcs have fixed (non-learned) large capacities and large costs. They therefore guarantee feasibility but are discouraged by their high costs, so the learned model eventually shifts away from using them.

**Other hyperparameters.** The depth and widths $(H, \ell_1, \ldots, \ell_H)$, as well as the affine scaling parameters $(\alpha, \beta)$, are selected using a validation set.

### 3.2. Learnable Parameters

We learn edge costs $c \in \mathbb{R}^m$, capacities $u \in \mathbb{Z}_{\geq 0}^m$, the matrix $A \in \mathbb{Z}^{p \times d}$ and the vector $b \in \mathbb{Z}^p$.

**Differentiation through min-cost flow.** For each training example $x$, we solve the associated minimum-cost flow problem (1) and obtain an optimal primal solution $\xi^\star$ along with optimal dual variables for the net outflow and capacity constraints. These solutions provide subgradients of $f(x; \Pi)$ with respect to the model parameters, which we use in standard stochastic optimization with Adam.

**Learning integer parameters via straight-through estimation (STE).** Because $u$, $A$, and $b$ are constrained to be integer-valued, we use straight-through estimation (STE) (Bengio et al., 2013). Moreover, since capacities must satisfy $u \geq 0$, we apply a softplus map to enforce nonnegativity. Concretely, we maintain real-valued parameters $(\bar{u}, \bar{A}, \bar{b})$. In the forward pass, we form the integer parameters used by the min-cost flow solver as

$$A = \text{round}(\bar{A}), \ b = \text{round}(\bar{b}), \ u = \text{round}(\text{softplus}(\bar{u})).$$

In the backward pass, STE treats the rounding operator $\text{round}(\cdot)$ as the identity map, allowing gradients to propagate to $(\bar{u}, \bar{A}, \bar{b})$.

**Initialization and keeping $\Delta(A)$ small.** As shown in Section 2.3, the complexity of optimizing a DFN function is closely related to the magnitude of $\Delta(A)$, which depends on the row corank of $A$ and its pseudo-determinant. To encourage small $\Delta(A)$, we keep the number of rows of $A$ small (while ensuring good learning performance) and initialize $A$ using a "near-identity" construction that cycles through the standard basis vectors in round-robin order: the rows of $A$ are set to $e_1, e_2, \ldots, e_d, e_1, e_2, \ldots$ until all $p$ rows are filled. For the remaining parameters, we initialize $b$ to $\mathbf{0}$, capacities to $u = \mathbf{1}$, and costs $c$ to i.i.d. $\mathcal{N}(1, 1)$.

## 4. Experiments

We evaluate DFN approximators on data generated by convex-extendible discrete functions, assessing (i) generalization on unseen data and (ii) the efficiency of integer optimization over the trained models.

### 4.1. Datasets

We generate supervised datasets $\mathcal{D} = \{(x_t, y_t)\}_{t=1}^{K}$ with integer-valued features $x_t$ and real-valued labels $y_t = f(x_t)$, where $f$ is a convex-extendible ground-truth function. We consider three datasets, drawn from qualitatively different generative processes to illustrate that DFN functions generalize well across different settings.

**Convex quadratic dataset.** We generate a dataset from a convex quadratic function. Each input $x_t \in \mathbb{Z}^d$ is sampled i.i.d. from the discrete uniform distribution over the integer box $[x_{\min}, x_{\max}]^d \cap \mathbb{Z}^d$. We sample a minimizer $x^\star \in$

$\mathbb{Z}^d$ uniformly at random from the same domain. We then construct a symmetric positive semidefinite matrix $Q \in \mathbb{R}^{d \times d}$ with eigenvalues $\{\lambda_j\}_{j=1}^d$ drawn independently from $\mathrm{Unif}[\lambda_{\min}, \lambda_{\max}]$ and a random orthonormal eigenbasis $U$ as,

$$Q = U \, \mathrm{diag}(\lambda_1, \ldots, \lambda_d) \, U^\top.$$

Targets are computed as

$$y_t = (x_t - x^\star)^\top Q(x_t - x^\star), \qquad t \in [K].$$

We use $K = 2000$, $d = 16$, $x_{\min} = -50$, $x_{\max} = 50$, and $\lambda_{\min} = 1$, $\lambda_{\max} = 15$.

**Minimum-cost resource allocation dataset.** We consider a minimum-cost *resource allocation* problem. There are $n$ resource types and $n$ allocation targets. A cost matrix $C \in \mathbb{R}^{n \times n}$ is sampled with entries $C_{ij} \sim \mathrm{Unif}\{c_{\min}, \ldots, c_{\max}\}$, where $C_{ij}$ is the cost of allocating one unit of resource type $i$ to target $j$. For each instance $t \in [K]$, we sample an integer resource vector $x_t \in \{0, \ldots, x_{\max}\}^n$ uniformly at random, where $(x_t)_i$ is the available amount of resource type $i$. Each target $j$ can absorb at most $x_{\max}$ units. The label $y_t$ is the minimum total allocation cost:

$$y_t = \min_{z \in \mathbb{Z}_{\geq 0}^{n \times n}} \quad \sum_{i=1}^n \sum_{j=1}^n C_{ij} \, z_{ij}$$
$$\text{s.t.} \quad \sum_{j=1}^n z_{ij} = (x_t)_i \ \forall i \in [n],$$
$$\sum_{i=1}^n z_{ij} \leq x_{\max} \ \forall j \in [n].$$

We use $K = 2000$, $n = 16$, $c_{\min} = 1$, $c_{\max} = 1000$, and $x_{\max} = 50$.

**Multiple-depot vehicle scheduling dataset.** We consider a multi-depot vehicle scheduling problem (MDVSP) with $m$ depots and $n$ timetabled trips. Each trip $i$ has a start time $s_i$, start location $p_i$, end time $e_i$, and end location $q_i$, and travel times are given by a matrix $d(\cdot, \cdot)$. A vehicle starts at a depot, executes a time-feasible sequence of trips (a trip $j$ can follow $i$ only if $s_j \geq e_i + d(q_i, p_j)$), and returns to a depot. The objective is to minimize operational cost (deadheading travel and waiting) while respecting vehicle availability at each depot.

We use public MDVSP benchmark instances from Kulkarni et al. and Mathprog-ORlib (Kulkarni et al., 2019). Concretely, we fix an instance of the dataset. We generate $K$ inputs by sampling depot vehicle availabilities $x_t \in \mathbb{Z}_{\geq 0}^m$ uniformly (component-wise) from $[0, x_{\max}]$. For each sampled $x_k$, we construct the standard MDVSP min-cost flow network with a super-source and super-sink, depot boundary

arcs whose capacities are set to $x_t$ (available vehicles per depot), unit-capacity trip arcs, depot-to-trip (pull-out) and trip-to-depot (pull-in) arcs, and feasible successor arcs between trips. Arc costs follow the benchmark definitions: pull-out $5000 + 10 \, d(\text{depot}, p_i)$, pull-in $5000 + 10 \, d(q_i, \text{depot})$, and trip-connection cost $8 \, d(q_i, p_j) + 2(s_j - e_i)$ for feasible successors (Kulkarni et al., 2019). We compute the maximum feasible flow value $F_{\max}$ under capacities $x_k$, then solve a minimum-cost flow of value $F_{\max}$. The resulting optimal cost is used as $y_k$[1].

We use $K = 2000$, $m = 16$, $n = 2000$, and $x_{\max} = 50$.

### 4.2. Implementation Details and Benchmarks

**Train/validation/test protocol and loss.** For each dataset we split $\mathcal{D}$ into training/validation/test sets. We standardize features and targets using the training set (to zero mean and unit variance), train by minimizing mean-squared error (MSE) on standardized targets, and report *normalized* MSE on test sets.

**Downstream integer optimization.** After training, we test the quality of integer optimization of each learned model via the following simple *budget-constrained* integer optimization problem:

$$\min_{x \in X} \quad \widehat{f}(x),$$
$$s.t. \quad X = \big\{x \in \mathbb{Z}^d \, : \, x_{\min} \leq x \leq x_{\max}, \ \mathbf{1}^\top x = B\big\},$$

where $\widehat{f}$ is the trained model and $B$ is a fixed budget.

**DFN.** The specifics of training and optimization of our DFN models are as follows:

- *Training.* We use a DFN with three layers and layer sizes $\ell_1 = 32$, $\ell_2 = 100$, $\ell_3 = 32$ (about $2.6 \times 10^4$ learnable parameters). The model is implemented in PyTorch. To evaluate $f(x; \Pi)$ in the forward pass, we solve the associated minimum-cost flow problem using the LEMON library (v1.3.1),

  We train with Adam for 1000 epochs with batch size 8 and learning rate $10^{-2}$. The affine scaling parameters $(\alpha, \beta)$ are selected by validation and set to $(0.005, -2)$.

- *Optimization.* While the local descent Algorithm 1 provides a theoretical framework linking the complexity of minimizing DFN functions to the quantity $\Delta(A)$,

---

[1]We note that although the mapping producing $y_k$ is not itself convex-extendible (we first compute a maximum flow and then, conditional on that value, minimize cost) the resulting dataset is still consistent with some convex-extendible generative function. In particular, after generating the data we verify that there exists a convex-extendible function that could have produced it.

motivating our initialization for the matrix $A$; we observe in practice that minimizing the learned DFN is often fastest when solved through its equivalent integer programming (IP) formulation. We therefore use this method for optimization of DFN functions in our experiments. Specifically, minimizing a trained DFN over a feasible set $X$ can be cast as the integer program

$$\min_{x,\xi} c^\top \xi$$

$$\text{s.t.} \sum_{e \in \delta^+_{v_i}} \xi_e - \sum_{e \in \delta^-_{v_i}} \xi_e = a_i^\top x + b_i, \ \forall i \in [p], \quad (2a)$$

$$\sum_{e \in \delta^+_v} \xi_e - \sum_{e \in \delta^-_v} \xi_e = 0, \quad \forall v \in V \setminus S, \quad (2b)$$

$$0 \le \xi_e \le u_e, \ \xi_e \in \mathbb{Z}, \quad \forall e \in E, \quad (2c)$$

$$x \in X. \quad (2d)$$

Intuitively, this integer program is a lot more structured than the IPs induced by ReLU MLP or LSET surrogates. Its constraint matrix is the concatenation of a network matrix (which is totally unimodular) and the matrix $A$. When most of the learning happens in costs and capacities and $A$ remains sufficiently simple the resulting program tends to be easier to optimize. We solve (2) using Gurobi.

**MLP.** Our first benchmark is a multi-layer perceptron (MLP) with ReLU activation.

- *Training.* We use a two-hidden-layer MLP with hidden widths 150 and ReLU activations, yielding a parameter count comparable to DFN. We train with Adam for 1000 epochs with batch size 8 and learning rate $10^{-3}$.

- *Optimization.* Optimizing a trained MLP ReLU network over integer inputs can be cast as a mixed-integer linear program (MILP) by encoding each ReLU with a standard big-$M$ formulation which we solve using Gurobi; see, e.g., (Fischetti & Jo, 2018).

**LSET.** We also compare against the LogSumExp transform (LSET) model of (Calafiore et al., 2019), a convex smooth approximation to a maximum of affine functions. Given temperature $T > 0$ and $N$ affine pieces $(a_k, b_k)$, the model is

$$\widehat{f}(x) \ = \ T \log\Big( \sum_{k=1}^N \exp\big((a_k^\top x + b_k)/T\big) \Big).$$

- *Training.* We use $N = 1570$ affine pieces and $T = 0.05$ (yielding a parameter count on the same order as DFN and MLP), and train with Adam for 1000 epochs with batch size 8 and learning rate $10^{-3}$.

*Table 1.* Generalization performance (normalized test MSE). Results are mean $\pm$ standard error over 5 seeds.

| Model | Quad ($\times 10^{-3}$) | RA ($\times 10^{-4}$) | MDVSP ($\times 10^{-6}$) |
|---|---|---|---|
| DFN | $3.4 \pm 0.4$ | $5.3 \pm 1.4$ | $4.7 \pm 2.5$ |
| MLP | $12.5 \pm 0.9$ | $8.0 \pm 2.8$ | $22.1 \pm 4.2$ |
| LSET | $80.4 \pm 7.5$ | $10.4 \pm 1.4$ | $9.1 \pm 1.7$ |

- *Optimization.* To optimize the trained LSET surrogate, we solve a mixed-integer convex program using Gurobi which supports general constraints involving $\exp(\cdot)$ and $\log(\cdot)$.

**Software and hardware.** All experiments were conducted on a MacBook Pro with an Apple M4 Max chip and 36 GB of RAM, using CPU only. We used PyTorch v2.8.0, Gurobi v12.0.3, and LEMON v1.3.1. Code is available at https://github.com/ayfous/deep-flow-networks.

### 4.3. Results

Results are reported as mean $\pm$ standard error over 5 random seeds. Each seed affects both data generation and training: for the quadratic dataset we resample the minimizer $x^\star$ and the matrix $Q$ (via its eigenvalues and eigenbasis); for the resource allocation dataset we resample the cost matrix $C$ and the input vectors $x_t$; and for the MDVSP dataset we resample the depot availability vectors $x_t$. Training is also stochastic due to minibatch sampling and Adam. Table 1 reports generalization performance in terms of normalized test MSE. Table 2 reports downstream optimization performance, including solver runtime in seconds (capped at 3600s with best incumbent reported at timeout) and the ratio between the objective value obtained by optimizing the learned model and the true optimum of the ground-truth function.

The results show that the DFN model achieves the lowest normalized test MSE across all datasets considered. Moreover, the associated integer program solves significantly faster than the integer programs for the MLP and LSET models, while still producing near-optimal solutions across all datasets. Overall, these findings suggest that DFN delivers strong predictive performance and enables efficient inference-time optimization, making it a compelling choice for learning from integer-valued data, particularly when downstream optimization is important.

*Table 2.* Downstream optimization over the trained model. Results are mean $\pm$ standard error over 5 seeds.

| Model | Quad | RA | MDVSP |
|---|---|---|---|
| | Time Sol/Opt | Time Sol/Opt | Time Sol/Opt |
| DFN | $1.8 \pm 0.4$ $4.9 \pm 1.0$ | $0.3 \pm 0.1$ $1.6 \pm 0.2$ | $0.1 \pm 0.0$ $1.0 \pm 0.0$ |
| MLP | $49.6 \pm 20.8$ $34.9 \pm 15.5$ | $16.1 \pm 4.5$ $1.1 \pm 0.0$ | $4.1 \pm 0.5$ $1.0 \pm 0.0$ |
| LSET | $3600 \pm 0.0$ $35.5 \pm 12.6$ | $3600.0 \pm 0.0$ $1.3 \pm 0.1$ | $284.8 \pm 96.2$ $1.0 \pm 0.0$ |

### 4.4. Additional Experiments

We report several additional experiments in Appendix B. These examine the evolution of the imbalance matrix $A$ during training and its relation to downstream optimization difficulty, the sensitivity of learning to initialization, more detailed results on the quadratic setting across different scales, and robustness to noisy labels.

## 5. Conclusion and Future Directions

We introduced the *Deep Flow Networks* (DFNs) approximators. We proved DFNs are universal approximators for convex-extendible discrete functions and connected the complexity of minimizing a learned DFN to its deviation from the $M$-convex regime through $\Delta(A)$. Empirically, DFNs achieve strong predictive performance while remaining substantially easier to optimize over integer inputs than standard neural baselines.

While our focus in this work is on generalization and optimization of trained models, improving DFN training time is a natural next step. Promising directions include (i) solving min-cost flow instances in batches, (ii) developing GPU-friendly min-cost flow implementations, and (iii) using approximate min-cost flow algorithms in place of exact algorithms during training. Another direction is to explore alternative methods for controlling $\Delta(A)$ during training, for instance, through explicit regularization of $\Delta(A)$ by penalizing an efficiently computable and differentiable surrogate or by freezing $A$ altogether and selecting it via validation from a fixed set of structured candidate matrices. A further direction is to evaluate DFNs on ground-truth objectives that are not necessarily convex-extendible. Since any function can be approximated as a difference of two convex functions, differences of two DFNs form a general approximation class for discrete functions. Understanding their empirical behavior, and how they compare to competing models, remains an important topic for future study.

## Impact Statement

Integer optimization is central to many real-world decision-making problems. A common pipeline uses machine learning to predict complex value functions, then integer programming to select decisions. Our paper provides a method to improve this pipeline, enabling faster decisions and better solutions in practice, which could have broad impact on domains such as supply chain logistics, energy grid management, and healthcare resource allocation.

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

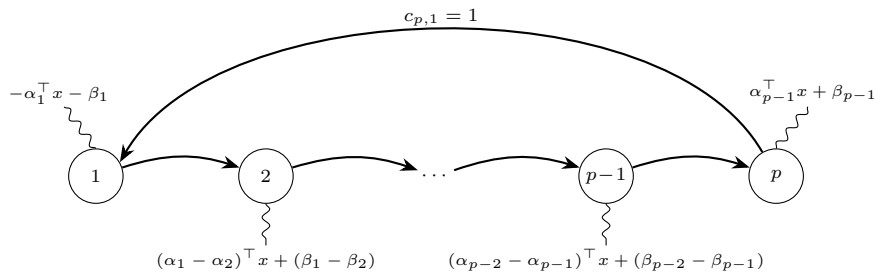

*Figure 1.* Construction of DFN $\Pi$ such that $f(x; \Pi) = \max\Big\{0,\ \max_{i=1,\dots,p-1}\big(\alpha_i^\top x + \beta_i\big)\Big\}$: Arc $(p, 1)$ has cost 1, and the unlabeled arcs have cost 0. Every arc has capacity $\max_{x \in I}(|\alpha_i^\top x| + |\beta_i|)$. All nodes are special nodes $S = \{1, \dots, p\}$ with net outflows $-\alpha_1^\top x - \beta_1, (\alpha_1 - \alpha_2)^\top x + (\beta_1 - \beta_2), \dots, (\alpha_{p-2} - \alpha_{p-1})^\top x + (\beta_{p-2} - \beta_{p-1}), \alpha_{p-1}^\top x + \beta_{p-1}$ respectively.

## A. Missing Proofs

### A.1. Proof of Theorem 2.1

Let $\alpha_1, \dots, \alpha_{p-1} \in \mathbb{Z}^d, \beta_1, \dots, \beta_{p-1} \in \mathbb{Z}$, and let $I \subset \mathbb{Z}^d$ be a finite set of integers. We first show that there exists a DFN $\Pi$ such that $f(x; \Pi) = \max\Big\{0,\ \max_{i=1,\dots,p-1}\big(\alpha_i^\top x + \beta_i\big)\Big\}$ for every $x \in I$.

Consider the DFN in Fig. 1. Fix $x \in I$, and let $\xi$ be any feasible solution of (1). For every $i = 2, \dots, p$, define $I_i := \{i, \dots, p\}$. By construction, the only arc leaving $I_i$ is $(p, 1)$ and the only arc entering $I_i$ is $(i - 1, i)$. Summing the net outflow constraints (1a) over all nodes in $I_i$ yields

$$\xi_{p,1} - \xi_{i-1,i} = \sum_{j=i}^{p-1}(\alpha_{j-1} - \alpha_j)^\top x + (\beta_{j-1} - \beta_j)\ +\ \alpha_{p-1}^\top x\ +\ \beta_{p-1} = \alpha_{i-1}^\top x + \beta_{i-1}.$$

Since $\xi_{i-1,i} \geq 0$, we obtain $\xi_{p,1} \geq \alpha_{i-1}^\top x + \beta_{k-1}$ for all $k = 2, \dots, p$, i.e., $\xi_{p,1} \geq \max_{i=1,\dots,p-1}\big(\alpha_i^\top x + \beta_i\big)$. Moreover $\xi_{p,1} \geq 0$. Because the unique positive-cost arc is $(p, 1)$ with $c_{p,1} = 1$ and all other arcs have zero cost, we have $c^\top \xi = \xi_{p,1}$ and therefore

$$f(x; \Pi)\ \geq\ \max\Big\{0,\ \max_{i=1,\dots,p-1}\big(\alpha_i^\top x + \beta_i\big)\Big\}.$$

For the reverse inequality, define $M := \max\Big\{0,\ \max_{i=1,\dots,p-1}\big(\alpha_i^\top x + \beta_i\big)\Big\}$. Set $\xi_{p,1} := M$ and, for each $i = 2, \dots, p$, set $\xi_{i-1,i} := M - (\alpha_{i-1}^\top x + \beta_{i-1}) \geq 0$. With these choices, the constraints (1a)–(1c) are all satisfied, hence $\xi$ is feasible, and has objective value $c^\top \xi = \xi_{p,1} = M$. Therefore $f(x; \Pi) \leq \max\Big\{0,\ \max_{i=1,\dots,p-1}\big(\alpha_i^\top x + \beta_i\big)\Big\}$.

Next, consider a convex function $g : \mathbb{R}^d \to \mathbb{R} \cup \{+\infty\}$, let $I = \{x^1, \dots, x^N\} \subset \operatorname{dom} g \cap \mathbb{Z}^d$ and $\epsilon > 0$. For each $i$, pick any subgradient $s^i \in \partial g(x^i)$. Define the supporting affine functions

$$\ell_i(x)\ :=\ g(x^i) + (s^i)^\top(x - x^i)\ =\ (s^i)^\top x + \Big(g(x^i) - (s^i)^\top x^i\Big), \qquad i = 1, \dots, N.$$

By convexity, $\ell_i(x) \leq g(x)$ for all $x \in \mathbb{R}^d$, and $\ell_i(x^i) = g(x^i)$. Let $h(x) := \max_{i=1,\dots,N} \ell_i(x)$. Then $h(x) \leq g(x)$ for all $x$, and for each $i \in \{1, \dots, N\}, h(x^i) \geq \ell_i(x^i) = g(x^i)$, while $h(x^i) \leq g(x^i)$, hence $h(x) = g(x)$ for every $x \in I$.

Note that $g(x) - \min_{x \in I} g(x) \geq 0$. Hence, for every $x \in I$,

$$g(x) - \min_{x \in I} g(x)\ =\ h(x) - \min_{x \in I} g(x)\ =\ \max\Big\{0,\ h(x) - \min_{x \in I} g(x)\Big\}\ =\ \max\Big\{0,\ \max_{i=1,\dots,N}\big(\ell_i(x) - \min_{x \in I} g(x)\big)\Big\}.$$

Since $\mathbb{Q}$ is dense in $\mathbb{R}$, there exists $\{q^i\}_i, \{r^i\}_i \in \mathbb{Q}^d$ such that,

$$\max_{x \in S} \left| g(x) - \left(\max\Big\{0,\ \max_{i=1,\dots,N}(q^i)^\top x + r^i\Big\} + \min_{x \in I} g(x)\right) \right| < \epsilon$$

Letting $\alpha$ be the inverse of the smallest common denominator of $q^i$ and $r^i$ for every $i$, $\beta = \min_{x \in I} g(x)$, and using the first part of the proof yeilds the result.

## A.2. Proof of Lemma 2.4

In an iteration of the while loop where an improvement $\tilde{x} \neq 0$ is found, if $\zeta \in \mathbb{Z}_{\geq 0}^E$ and $\zeta' \in \mathbb{Z}_{\geq 0}^E$ denote optimal solutions of the minimum cost flow integer programs given by $f(x; \Pi)$ and $f(x + \tilde{x}; \Pi)$ respectively, then the (strict) improvement in the cost is given by, $c^\top(\zeta - \zeta') \geq \frac{1}{D}$. Moreover, the maximum cost of any feasible $\zeta$ is at most $n^2 \|u\|_\infty \|c\|_\infty$, implying that the algorithm finishes in at most $n^2 \|u\|_\infty \|c\|_\infty D$ iterations of the while loop.

Every iteration of the while loop enumerates $O(p^{\Delta(A)})$ integer vectors $y \in \mathbb{Z}^p$ such that $\|y\|_1 \leq \Delta(A)$, checks whether the system of Diophantine equations $A\tilde{x} = y$ has a solution which can be done in time polynomial in $p, \log(\|A\|_\infty)$ and $\log(\|y\|_\infty) \leq \log(\Delta(A))$ (see. e.g. (Schrijver, 1986)) and checks if $f(x + \tilde{x}; \Pi) < f(x; \Pi)$ which can also be done in time polynomial in $n$ (see. e.g. (Tardos, 1985)).

When no improvement is found and the algorithm finishes, the returned solution $x$ is an optimal solution. In fact, assume not, and let $x^*$ denote an optimal solution such that $\|Ax^* - Ax\|_1$ is minimal. We have $Ax \neq Ax^*$ (otherwise $f(x; \Pi) = f(x^*; \Pi)$) and hence, by the generalized exchange property (G-EXC), there exists $\tilde{x}$ with $0 < \|A\tilde{x}\|_1 \leq \Delta(A)$, $(A\tilde{x})^+ \leq (Ax^* - Ax)^+$ and $(A\tilde{x})^- \leq (Ax^* - Ax)^+$ such that,

$$f(x^*; \Pi) + f(x; \Pi) \geq f(x^* - \tilde{x}; \Pi) + f(x + \tilde{x}; \Pi)$$

Since $f(x; \Pi) \leq f(x + \tilde{x}; \Pi)$ (otherwise an improvement would have been found when $y = A\tilde{x}$ was examined in the for loop), and $f(x^*; \Pi) \leq f(x^* - \tilde{x}; \Pi)$ ($x^*$ is a global minimum) it must be the case that $f(x^* - \tilde{x}; \Pi^*) = f(x^*; \Pi^*)$ and hence that $x^* - \tilde{x}$ is also an optimal solution, however,

$$\|(A(x^* - \tilde{x}) - Ax\|_1 = \|(Ax^* - Ax) - A\tilde{x}\|_1 = \|Ax^* - Ax\|_1 - \|A\tilde{x}\|_1 < \|Ax^* - Ax\|_1$$

where the second equality is because $(A\tilde{x})^+ \leq (Ax^* - Ax)^+$ and $(A\tilde{x})^- \leq (Ax^* - Ax)^+$. A contradiction to minimality of $\|Ax^* - Ax\|_1$.

## A.3. Proof of Lemma 2.5

A feasible $s$-$t$ flow in a directed multi-graph $\hat{G} = (\hat{V}, \hat{E})$ with source node $s$, sink node $t$, and arc capacities $\hat{u}$ is a vector $\hat{\Delta} \in [0, \hat{u}]$ such that

$$\sum_{e \in \delta_v^+} \hat{\Delta}_e - \sum_{e \in \delta_v^-} \hat{\Delta}_e = 0 \quad \forall v \in \hat{V} \setminus \{s, t\}, \qquad \text{and} \qquad \sum_{e \in \delta_s^+} \hat{\Delta}_e - \sum_{e \in \delta_s^-} \hat{\Delta}_e = -\left( \sum_{e \in \delta_t^+} \hat{\Delta}_e - \sum_{e \in \delta_t^-} \hat{\Delta}_e \right).$$

A well known result (a.k.a flow decomposition theorem) states that any integer feasible $s$-$t$ flow can be decomposed as a sum of unit $s$-$t$ flows supported on either a directed $s$-$t$ path or a directed cycle; see, e.g., (Ahuja et al., 1993).

Consider $y, y' \in \text{dom} f(\cdot; \Pi^*)$ with $y \neq y'$, and let $\xi, \xi' \in \mathbb{Z}_+^m$ be optimal solutions of the minimum-cost flow integer programs defining $f(y; \Pi^*)$ and $f(y'; \Pi^*)$, respectively. Let $d := y - y'$ and $\Delta := \xi - \xi' \in \mathbb{Z}^m$.

**Auxiliary $s$-$t$ network.** Create a directed multi-graph $\hat{G}$ on node set $V \cup \{s, t\}$ as follows. First, for each original arc $e = (u, v) \in E$, add $|\Delta_e|$ parallel unit-capacity copies, oriented as:

- $(u, v)$ if $\Delta_e > 0$,
- $(v, u)$ if $\Delta_e < 0$.

Second, add $(d_i)^+$ parallel unit-capacity arcs $(s, v_i)$ and $(d_i)^-$ parallel unit-capacity arcs $(v_i, t)$ for every $i \in [p]$.

Define $\hat{\Delta}$ as the integer $s$-$t$ flow on $\hat{G}$ that sends one unit of flow on every arc of $\hat{G}$. By construction, for each $i \in [p]$, the contribution of the arc-copies coming from $\Delta$ gives net outflow $d_i$ at $v_i$ (whereas all other $v \in V \setminus \{v_1, \ldots, v_p\}$ have net outflow 0), and the added arcs $(s, v_i)$ and $(v_i, t)$ contribute net outflow $-(d_i)^+ + (d_i)^- = -d_i$ at $v_i$. Hence the total net outflow at every $v \in V$ is 0, while

$$\sum_{e \in \delta_s^+} \hat{\Delta}_e - \sum_{e \in \delta_s^-} \hat{\Delta}_e = \sum_{i=1}^p (d_i)^+ =: K, \qquad \text{and} \qquad \sum_{e \in \delta_t^+} \hat{\Delta}_e - \sum_{e \in \delta_t^-} \hat{\Delta}_e = -\sum_{i=1}^p (d_i)^- = -K.$$

Therefore $\hat{\Delta}$ is a feasible integer $s$-$t$ flow of value $K$ in $\hat{G}$.

**Flow decomposition and pairing.** By the flow decomposition theorem (Ahuja et al., 1993), we can write

$$\hat{\Delta} = \sum_{k \in P} \Delta^k + \sum_{k \in C} \Delta^k,$$

where each $\Delta^k$ is a unit $s$-$t$ flow supported on a directed $s$-$t$ path if $k \in P$, and supported on a directed cycle if $k \in C$. Since $s$ has only outgoing arcs and $t$ has only incoming arcs in $\hat{G}$, no directed cycle can include $s$ or $t$; hence the entire $s$-$t$ flow value $K$ is carried by the path flows, i.e., $|P| = K$.

Index the $K$ path flows as $P = \{1, \ldots, K\}$. Each unit $s$-$t$ path $\Delta^k$ must start with some arc $(s, v_{i_k})$ and end with some arc $(v_{j_k}, t)$. Because there are exactly $(d_i)^+$ parallel arcs leaving $s$ toward $v_i$ (each carrying one unit), the multiset $\{i_1, \ldots, i_K\}$ contains each $i$ exactly $(d_i)^+$ times; similarly, $\{j_1, \ldots, j_K\}$ contains each $j$ exactly $(d_j)^-$ times. Thus,

$$d^+ = (y - y')^+ = \sum_{k=1}^{K} e_{i_k}, \qquad \text{and} \qquad d^- = (y - y')^- = \sum_{k=1}^{K} e_{j_k}.$$

Define $\tilde{y}_k := e_{i_k} - e_{j_k}$ and, for any $T \subseteq [K]$, $\tilde{y}_T := \sum_{k \in T} \tilde{y}_k$.

**Constructing feasible flows.** Fix $T \subseteq [K]$. For each $k \in T$, consider the internal portion of the $s$-$t$ path $\Delta^k$ between $v_{i_k}$ and $v_{j_k}$. Define an adjustment vector $\eta^k \in \{-1, 0, 1\}^E$ by setting, for each original arc $e = (u, v) \in E$,

$$\eta_e^k := \begin{cases} -1, & \text{if the path uses a copy oriented as } (u, v) \text{ (equivalently } \Delta_e > 0), \\ +1, & \text{if the path uses a copy oriented as } (v, u) \text{ (equivalently } \Delta_e < 0), \\ 0, & \text{otherwise.} \end{cases}$$

Now define

$$\xi^T := \xi + \sum_{k \in T} \eta^k, \qquad \text{and} \qquad (\xi')^T := \xi' - \sum_{k \in T} \eta^k.$$

Because the path decomposition uses each arc-copy at most once and there are $|\Delta_e|$ copies attached to each $e$, the update above modifies $\xi_e$ and $\xi'_e$ by at most $|\Delta_e| = |\xi_e - \xi'_e|$ units on each arc $e$, and in the direction from the larger to the smaller, so $0 \le \xi_e^T, (\xi')_e^T \le u_e$ for all $e \in E$. Moreover, along the internal nodes of each path the $+1/-1$ changes cancel in the flow-balance constraints, and only the endpoints $v_{i_k}, v_{j_k}$ experience a net change: $\xi^T$ decreases the net supply at $i_k$ by 1 and increases the net supply at $j_k$ by 1. Hence $\xi^T$ is feasible for the right-hand side $y - \tilde{y}_T$ and $(\xi')^T$ is feasible for $y' + \tilde{y}_T$ in (1).

**Cost comparison and conclusion.** Finally, by construction the arc-wise changes cancel in the sum of costs:

$$c^\top \xi^T + c^\top (\xi')^T = c^\top \xi + c^\top \xi'.$$

Since $\xi^T$ and $(\xi')^T$ are feasible for $y - \tilde{y}_T$ and $y' + \tilde{y}_T$, respectively, we have

$$f(y - \tilde{y}_T; \Pi^*) + f(y' + \tilde{y}_T; \Pi^*) \le c^\top \xi^T + c^\top (\xi')^T = c^\top \xi + c^\top \xi' = f(y; \Pi^*) + f(y'; \Pi^*),$$

which proves the desired inequality.

### A.4. Proof of Lemma 2.9

**Reminder of basic lattice theory notions and results.**

**Definition A.1** (Lattice). A lattice $\mathcal{L} \subset \mathbb{Z}^p$ is a subgroup of $\mathbb{Z}^p$. That is, for every $x, y \in \mathcal{L}$, $x - y \in \mathcal{L}$.

**Definition A.2** ($\mathbb{Z}$-basis). Given a latt ice $\mathcal{L}$. A $\mathbb{Z}$-basis of $\mathcal{L}$ is a set of vectors $b_1, \ldots, b_k \in \mathcal{L}$ that are linearly independent and such that every $y \in \mathcal{L}$ can be written as $y = \lambda_1 b_1 + \cdots + \lambda_k b_k$ for some $\lambda_1, \ldots, \lambda_k \in \mathbb{Z}$. The *row-stacked* basis matrix

$$B := \begin{pmatrix} b_1^\top \\ \vdots \\ b_k^\top \end{pmatrix} \in \mathbb{Z}^{k \times p}.$$

is called a $\mathbb{Z}$-basis matrix of $\mathcal{L}$.

**Definition A.3** (Orthogonality Defect). Given a lattice $\mathcal{L}$, and a $\mathbb{Z}$-basis $b_1, \ldots, b_k$. Let $b_1^*, \ldots, b_k^*$ denote the Gram-Schmidt orthogonalization of $b_1, \ldots, b_k$. The orthogonality defect of $b_1, \ldots, b_k$ is given by,

$$\gamma(b_1, \ldots, b_k) = \prod_{i=1}^{k} \frac{\|b_i\|_2}{\|b_i^*\|_2}$$

**Lemma A.4** (Gram determinant identity; see, e.g., (Galbraith, 2012)). *Given any linearly independent vectors $b_1, \ldots, b_k$, such that $b_1^*, \ldots, b_k^*$ is their Gram–Schmidt orthogonalization we have,*

$$\prod_{i=1}^{k} \|b_i^*\|_2 = \sqrt{\det(BB^\top)}.$$

*In particular, for every lattice $\mathcal{L}$ with $\mathbb{Z}$-basis $b_1, \ldots, b_k$, it holds that*

$$\prod_{i=1}^{k} \|b_i\|_2 = \gamma(b_1, \ldots, b_k) \cdot \sqrt{\det(BB^\top)}.$$

**Lemma A.5** (KZ-reduced basis; see, e.g., (Rothvoss, 2023; Lagarias et al., 1990)). *Given a lattice $\mathcal{L}$, there exists a $\mathbb{Z}$-basis $b_1, \ldots, b_k$ of $\mathcal{L}$ of orthogonality defect at most $k^k$. One such basis is the so-called Korkine–Zolotarev (KZ) reduced basis.*

**The last $k$ rows of $V^{-1}$ can be chosen to be any $\mathbb{Z}$-basis of $\ker_{\mathbb{Z}}(A^T)$.** Define

$$\ker_{\mathbb{Z}}(A^T) := \{y \in \mathbb{Z}^p : A^T y = 0\}.$$

This is a lattice (a subgroup of $\mathbb{Z}^p$). Let $W := V^{-1}$ and let $w_1^\top, \ldots, w_p^\top$ be the rows of $W$.

**Lemma A.6.** *The rows $w_{r+1}^\top, \ldots, w_p^\top$ form a $\mathbb{Z}$-basis of $\ker_{\mathbb{Z}}(A^T)$. Moreover, if $\tilde{w}_1^\top, \ldots, \tilde{w}_k^\top$ is any other $\mathbb{Z}$-basis of $\ker_{\mathbb{Z}}(A^T)$, then there exists another SNF decomposition $A = V'DU$ such that the last $k$ rows of $(V')^{-1}$ are exactly $\tilde{w}_1^\top, \ldots, \tilde{w}_k^\top$.*

*Proof. Step 1: the last $k$ rows are independent and lie in the integer kernel:*

For $\ell = 1, \ldots, k$, $A^T w_{r+\ell} = U^\top D^\top V^\top w_{r+\ell} = U^\top D^\top e_{r+\ell} = 0$, because the last $k$ rows of $D$ are zero. Thus $w_{r+\ell} \in \ker_{\mathbb{Z}}(A^T)$. They are linearly independent because $W$ is invertible.

*Step 2: the last $k$ rows generate $\ker_{\mathbb{Z}}(A^T)$ over $\mathbb{Z}$:*

Let $y \in \ker_{\mathbb{Z}}(A^T)$. Then $0 = y^\top A = y^\top VDU$, hence $(y^\top V)D = 0$. Set $\hat{y}^\top := y^\top V \in \mathbb{Z}^{1 \times p}$. Because $D$ has nonzero diagonal only in positions $1, \ldots, r$, the equation $\hat{y}^\top D = 0$ forces $\hat{y}_1 = \cdots = \hat{y}_r = 0$. Therefore $\hat{y}^\top$ is an integer combination of $e_{r+1}^\top, \ldots, e_p^\top$, and hence

$$y^\top = \hat{y}^\top V^{-1}$$

is an integer combination of the last $k$ rows of $V^{-1} = W$.

From Step 1 and 2, $w_{r+1}^\top, \ldots, w_p^\top$ is a $\mathbb{Z}$-basis of $\ker_{\mathbb{Z}}(A^T)$.

*Step 3: replacing the last $k$ rows by any other $\mathbb{Z}$-basis $\tilde{w}_1^\top, \ldots, \tilde{w}_k^\top$.*

Let $\hat{W} \in \mathbb{Z}^{k \times p}$ and $\tilde{W} \in \mathbb{Z}^{k \times p}$ be the row-stacked matrix with rows $w_{r+1}^\top, \ldots, w_p^\top$ and $\tilde{w}_1^\top, \ldots, \tilde{w}_k^\top$, respectively. Since both $\hat{W}$ and $\tilde{W}$ are $\mathbb{Z}$-basis matrices of the same lattice (hence can be written as linear integer combinations of each other), there is a unimodular matrix $P \in \mathbb{Z}^{k \times k}$ such that $\tilde{W} = P\hat{W}$. Define the block-diagonal unimodular matrix $S := \begin{pmatrix} I_r & 0 \\ 0 & P \end{pmatrix} \in \mathbb{Z}^{p \times p}$ and set

$$V' := VS^{-1}.$$

Since the last $k$ rows of $D$ are zero, one has $S^{-1}D = D$, hence

$$A = VDU = (VS^{-1})DU = V'DU,$$

so $A = V'DU$ is again an SNF of $A$. Finally $(V')^{-1} = SV^{-1}$, so its last $k$ rows are $P$ times the last $k$ rows of $V^{-1}$, i.e. $\tilde{W}$. $\square$

**A determinant bound.** The following lemma relates $\hat{W}\hat{W}^\top$ to $\det^*(A)$ and $\prod_{i=1}^r s_i$. Recall that $\hat{W}$ is the row-stacked matrix of the last $k$ rows of $W = V^{-1}$.

**Lemma A.7.**

$$\sqrt{\det(\hat{W}\hat{W}^\top)} \leq \frac{\det^*(A)}{\prod_{i=1}^r s_i}.$$

*Proof. Submatrix notation.* For a matrix $M \in \mathbb{R}^{m \times n}$ and index sets $I \subseteq [m]$, $J \subseteq [n]$, we write $M_{[I,J]}$ for the submatrix of $M$ obtained by restricting to rows in $I$ and columns in $J$. We also use the shorthand $M_{[I,:]} := M_{[I,[n]]}$ for the row-restriction and $M_{[:,J]} := M_{[[m],J]}$ for the column-restriction. If $I \subseteq [p]$, we denote by $I^c := [p] \setminus I$ its complement in $[p] := \{1, \ldots, p\}$.

Let $S := \mathrm{diag}_{r,r}(s_1, \ldots, s_r) \in \mathbb{Z}^{r \times r}$, so that

$$D = \begin{pmatrix} S & 0 \\ 0 & 0 \end{pmatrix} \in \mathbb{Z}^{p \times d}.$$

Let $\widetilde{U} := U_{[1..r,:]} \in \mathbb{Z}^{r \times d}$ be the matrix of the first $r$ rows of $U$, and let $\widetilde{V} := V_{[:,1..r]} \in \mathbb{Z}^{p \times r}$ be the matrix of the first $r$ columns of $V$. Then

$$A = VDU = \widetilde{V}\,S\,\widetilde{U}.$$

*Step 1:* $\det(\hat{W}\hat{W}^\top) = \det(\widetilde{V}^\top\widetilde{V})$.

Let $I := \{r+1, \ldots, p\}$. By the Cauchy–Binet formula (Horn & Johnson, 2012),

$$\det(\hat{W}\hat{W}^\top) = \sum_{\substack{J \subseteq [p] \\ |J|=k}} \det\big((V^{-1})_{[I,J]}\big)^2.$$

By Jacobi's complementary minor identity (Horn & Johnson, 2012), for every $J \subseteq [p]$ with $|J| = |I| = k$,

$$\det\big((V^{-1})_{[I,J]}\big) = \frac{\pm\det\big(V_{[J^c, I^c]}\big)}{\det(V)}.$$

Since $V$ is unimodular, $\det(V) = \pm 1$, and since $I^c = \{1, \ldots, r\}$, we have $V_{[J^c, I^c]} = \widetilde{V}_{[J^c,:]}$. Therefore,

$$\det\big((V^{-1})_{[I,J]}\big)^2 = \det\big(\widetilde{V}_{[J^c,:]}\big)^2.$$

As $J$ ranges over all $k$-subsets of $[p]$, the complements $J^c$ range over all $r$-subsets of $[p]$. Hence

$$\det(\hat{W}\hat{W}^\top) = \sum_{\substack{I' \subseteq [p] \\ |I'|=r}} \det\big(\widetilde{V}_{[I',:]}\big)^2.$$

Applying Cauchy–Binet again gives

$$\det(\widetilde{V}^\top\widetilde{V}) = \sum_{\substack{I' \subseteq [p] \\ |I'|=r}} \det\big(\widetilde{V}_{[I',:]}\big)^2,$$

so indeed $\det(\hat{W}\hat{W}^\top) = \det(\widetilde{V}^\top\widetilde{V})$.

*Step 2: comparing* $\det^*(A)$ *to* $\det(S)\sqrt{\det(\widetilde{V}^\top\widetilde{V})}$.

Let $\lambda_1, \ldots, \lambda_r > 0$ be the nonzero eigenvalues of $AA^\top$. By definition, $\det^*(A)^2 = \det^*(AA^\top) = \prod_{i=1}^r \lambda_i$. For a symmetric $p \times p$ matrix of rank $r$, $\prod_{i=1}^r \lambda_i$ equals the sum of all $r \times r$ principal minors; hence

$$\det^*(A)^2 = \sum_{\substack{I' \subseteq [p] \\ |I'|=r}} \det\big((AA^\top)_{[I',I']}\big) = \sum_{|I'|=r} \det\big(A_{[I',:]}A_{[I',:]}^\top\big).$$

By Cauchy–Binet, for each such $I'$,

$$\det\big(A_{[I',:]}A_{[I',:]}^\top\big) = \sum_{\substack{J\subseteq[d]\\|J|=r}} \det\big(A_{[I',J]}\big)^2,$$

and therefore

$$\det^*(A)^2 = \sum_{|I'|=r}\sum_{|J|=r} \det\big(A_{[I',J]}\big)^2.$$

Using $A = \widetilde{V}\,S\,\widetilde{U}$, we have for all $|I'| = |J| = r$,

$$A_{[I',J]} = \widetilde{V}_{[I',:]}\,S\,(\widetilde{U})_{[:,J]},$$

so determinants factor:

$$\det\big(A_{[I',J]}\big) = \det\big(\widetilde{V}_{[I',:]}\big)\,\det(S)\,\det\big((\widetilde{U})_{[:,J]}\big).$$

Squaring and summing separates:

$$\det^*(A)^2 = \det(S)^2\Big(\sum_{|I'|=r}\det\big(\widetilde{V}_{[I',:]}\big)^2\Big)\Big(\sum_{|J|=r}\det\big((\widetilde{U})_{[:,J]}\big)^2\Big).$$

Since $\widetilde{U}$ has rank $r$, there exists some $J$ with $\det((\widetilde{U})_{[:,J]}) \neq 0$, and this determinant is an integer, hence $|\det((\widetilde{U})_{[:,J]})| \geq 1$. Thus

$$\sum_{|J|=r}\det\big((\widetilde{U})_{[:,J]}\big)^2 \geq 1,$$

and so

$$\det^*(A)^2 \geq \det(S)^2\sum_{|I'|=r}\det\big(\widetilde{V}_{[I',:]}\big)^2 = \det(S)^2\det(\widetilde{V}^\top\widetilde{V}).$$

Using Step 1 and $\det(S) = \prod_{i=1}^r s_i$, we conclude

$$\det^*(A)^2 \geq \Big(\prod_{i=1}^r s_i\Big)^2 \det(\hat{W}\hat{W}^\top),$$

and taking square-roots proves the claim:

$$\sqrt{\det(\hat{W}\hat{W}^\top)} \leq \frac{\det^*(A)}{\prod_{i=1}^r s_i}. \qquad\qquad \square$$

**Conclusion.** By Lemma A.6 and Lemma A.5, we may choose the SNF decomposition so that the last $k$ rows of $V^{-1}$ form a $\mathbb{Z}$-basis $w_{r+1}^\top,\ldots,w_p^\top$ of $\ker_{\mathbb{Z}}(A^T)$ with $\gamma(w_{r+1},\ldots,w_p) \leq k^k$. Then

$$\prod_{i=r+1}^p \max_{j=1,\ldots,p}\big|(V^{-1})_{ij}\big| = \prod_{i=r+1}^p \|w_i\|_\infty$$

$$\leq \prod_{i=r+1}^p \|w_i\|_2$$

$$= \gamma(\hat{v}_1,\ldots,\hat{v}_k)\cdot\sqrt{\det(\hat{W}\hat{W}^\top)} \qquad \text{(Lemma A.4)}$$

$$\leq k^k\cdot\sqrt{\det(\hat{W}\hat{W}^\top)}$$

$$\leq k^k\cdot\frac{\det^*(A)}{\prod_{i=1}^r s_i} \qquad \text{(Lemma A.7)}.$$

implying the desired result. $\qquad\qquad\square$

## B. Additional Experiments

Unless noted otherwise, all experiments use the convex quadratic dataset setting from the main text, the same 1000-epoch training budget, and approximately parameter-matched baselines. All reported quantities are mean $\pm$ standard error over 3 seeds.

### B.1. Evolution of $A$ and Downstream Optimization Time

**Objective.** We study how downstream optimization difficulty changes as the learned matrix $A$ moves away from its identity-like initialization $A_0$, which is designed to keep DFN near the easy optimization regime.

**Setup.** We train the DFN used for the convex quadratic dataset setting in the main text and save checkpoints every 100 epochs. To isolate the effect of $A$, we take the final trained model, keep all non-$A$ parameters fixed, and replace only $A$ by the checkpoint value before solving the downstream optimization problem. We report $\|A - A_0\|_1$, $\log \operatorname{pdet}(A)$, and downstream solve time.

*Table 3.* Evolution of $A$ and downstream optimization time across training checkpoints.

| Epoch | $\|A - A_0\|_1$ | $\log \operatorname{pdet}(A)$ | Solve time (s) |
|---|---|---|---|
| 100 | $1.67 \pm 0.33$ | $11.34 \pm 0.09$ | $0.119 \pm 0.018$ |
| 200 | $110.67 \pm 34.89$ | $15.04 \pm 1.26$ | $0.181 \pm 0.010$ |
| 300 | $111.00 \pm 35.04$ | $15.04 \pm 1.26$ | $0.196 \pm 0.028$ |
| 400 | $186.33 \pm 16.76$ | $20.23 \pm 1.04$ | $0.206 \pm 0.031$ |
| 500 | $720.67 \pm 74.06$ | $33.35 \pm 1.55$ | $0.290 \pm 0.053$ |
| 600 | $1278.67 \pm 54.18$ | $41.30 \pm 0.96$ | $0.454 \pm 0.097$ |
| 700 | $1694.67 \pm 33.94$ | $45.37 \pm 0.77$ | $0.582 \pm 0.187$ |
| 800 | $2034.00 \pm 38.35$ | $48.04 \pm 0.73$ | $1.323 \pm 0.320$ |
| 900 | $2291.33 \pm 21.93$ | $49.88 \pm 0.52$ | $1.180 \pm 0.296$ |
| 1000 | $2506.33 \pm 29.08$ | $51.35 \pm 0.53$ | $3.350 \pm 1.788$ |

**Discussion.** Both measures of deviation increase over training, and downstream solve time rises with them. Solve time is strongly correlated with both $\|A - A_0\|_1$ and $\log \operatorname{pdet}(A)$, which is consistent with the theory that moving farther away from the identity-like regime makes optimization harder.

### B.2. Initialization Sensitivity

**Objective.** We study how sensitive learning is to the initialization used in the main text. In particular, we test whether DFN still trains reliably under more randomized initialization schemes for $(A, b)$, $c$, and $u$.

**Setup.** We compare the default initialization against three more randomized initialization schemes: $(A, b)$ uniform over $\{-5, \ldots, 5\}$, $c$ uniform over $[-5, 5]$, and $u$ uniform over $\{1, \ldots, 5\}$. We also include the parameter-matched MLP and LSET baselines for reference.

*Table 4.* Initialization sensitivity study.

| Initialization / Model | Test MSE |
|---|---|
| Default init | $0.0031 \pm 0.0005$ |
| $(A, b) \sim U(\{-5, \ldots, 5\})$ | $0.0019 \pm 0.0003$ |
| $c \sim U([-5, 5])$ | $0.0039 \pm 0.0001$ |
| $u \sim U(\{1, \ldots, 5\})$ | $0.0023 \pm 0.0001$ |
| MLP | $0.0132 \pm 0.0014$ |
| LSET | $0.0830 \pm 0.0047$ |

**Discussion.** All DFN variants train successfully, and all remain clearly more accurate than MLP and LSET. Across the different DFN initialization schemes, final test errors remain in the same low range, indicating that the method is not brittle to a particular choice of initialization. The initialization used in the main text remains our preferred choice, since it is motivated not only by predictive performance but also by keeping the model closer to the easy optimization regime.

### B.3. Additional Scales of Data and Model Size

**Objective.** We study how training time, downstream optimization time, and test MSE scale as the data dimension and model size increase. In particular, we evaluate larger quadratic instances while keeping model sizes approximately matched across DFN, a variant of DFN with $A$ fixed at its near-identity initialization (DFN-fixedA), MLP, and LSET.

**Setup.** We consider four scales: small ($d = 8, K = 1000$, approximately 5k parameters), medium ($d = 16, K = 2000$, approximately 25k parameters), large ($d = 24, K = 4000$, approximately 50k parameters), and xlarge ($d = 32, K = 6000$, approximately 100k parameters). Downstream solve times use a one-hour time limit.

*Table 5.* Scaling results across larger quadratic instances.

| Metric | Model | Small | Medium | Large | XLarge |
|---|---|---|---|---|---|
| Training time (s) | DFN | $108.8 \pm 8.9$ | $597.9 \pm 28.0$ | $2161.7 \pm 182.4$ | $6443.1 \pm 957.0$ |
| | DFN-fixedA | $122.6 \pm 3.4$ | $1005.0 \pm 3.9$ | $3347.5 \pm 27.8$ | $9424.3 \pm 168.6$ |
| | MLP | $27.8 \pm 1.0$ | $66.5 \pm 2.1$ | $248.1 \pm 35.5$ | $734.1 \pm 69.4$ |
| | LSET | $32.0 \pm 1.2$ | $89.1 \pm 3.8$ | $264.7 \pm 33.9$ | $672.1 \pm 77.4$ |
| Optimization time (s) | DFN | $0.068 \pm 0.011$ | $1.18 \pm 0.20$ | $15.74 \pm 1.57$ | $815.41 \pm 241.38$ |
| | DFN-fixedA | $0.050 \pm 0.008$ | $0.212 \pm 0.005$ | $0.570 \pm 0.118$ | $0.758 \pm 0.090$ |
| | MLP | $1.29 \pm 0.49$ | $55.01 \pm 35.42$ | $3563.43 \pm 36.87$ | $3600.48 \pm 0.04$ |
| | LSET | $1560.47 \pm 1034.97$ | $2656.78 \pm 943.83$ | $3600.60 \pm 0.23$ | $3165.12 \pm 435.39$ |
| Test MSE | DFN | $0.0108 \pm 0.0070$ | $0.0033 \pm 0.0005$ | $0.0040 \pm 0.0009$ | $0.0032 \pm 0.0002$ |
| | DFN-fixedA | $0.0563 \pm 0.0168$ | $0.0698 \pm 0.0015$ | $0.0734 \pm 0.0095$ | $0.0691 \pm 0.0054$ |
| | MLP | $0.0067 \pm 0.0010$ | $0.0127 \pm 0.0015$ | $0.0207 \pm 0.0009$ | $0.0164 \pm 0.0025$ |
| | LSET | $0.0366 \pm 0.0049$ | $0.0862 \pm 0.0081$ | $0.1417 \pm 0.0070$ | $0.1265 \pm 0.0143$ |

**Discussion.** DFN is about one order of magnitude slower to train than MLP and LSET, which is the main computational cost of our approach. However, across scales, DFN retains strong approximation quality and a large downstream optimization advantage. The fixed-$A$ variant is even faster to optimize, at the cost of lower approximation quality.

### B.4. Noise Robustness

**Objective.** We study whether DFN remains reliable when the labels are noisy. In particular, we measure how approximation quality degrades as observation noise is added to the target values.

**Setup.** We use the same quadratic dataset as in the main text and the same models. For each seed, we compute the clean-label standard deviation and inject Gaussian noise with standard deviation equal to $\{0, 0.05, 0.10, 0.20\}$ times that quantity. We then retrain DFN, MLP, and LSET under the same training budget as in the main text.

*Table 6.* Noise robustness study.

| Noise ratio | DFN MSE | MLP MSE | LSET MSE |
|---|---|---|---|
| 0.00 | $0.0031 \pm 0.0005$ | $0.0132 \pm 0.0014$ | $0.0830 \pm 0.0047$ |
| 0.05 | $0.0070 \pm 0.0006$ | $0.0167 \pm 0.0002$ | $0.0867 \pm 0.0046$ |
| 0.10 | $0.0176 \pm 0.0002$ | $0.0294 \pm 0.0004$ | $0.0962 \pm 0.0054$ |
| 0.20 | $0.0541 \pm 0.0009$ | $0.0749 \pm 0.0019$ | $0.1225 \pm 0.0078$ |

**Discussion.**    All models degrade smoothly as noise increases, with no failed DFN runs. DFN remains the most accurate model at every noise level, and the gap over MLP and LSET persists even at the highest noise level tested. This suggests that the training procedure is stable and that the architecture is robust to moderate label noise.

