# OpenReview forum: "Deep Flow Networks"
_ICML.cc/2026/Conference — ICML 2026 spotlight_

### Official Review · Reviewer_VBCU · 2026-03-05

**Soundness:** 2
**Presentation:** 1
**Significance:** 1
**Originality:** 2
**Overall Recommendation:** 4
**Confidence:** 3

**Summary:**

- This paper introduces Deep Flow Networks (DFN), which is a class of discrete function approximators that is a generalization of the minimum-cost flow value functions.
- Then, it provides a theoretical analysis and proof on the universality and minimization complexity of the proposed DFN in order to illustrate its practical utility.
- The paper concludes with a section on DFN implementations and a set of experiments showing the performance and efficiency over several different datasets and problem settings.

**Compliance With Llm Reviewing Policy:**

Affirmed.

**Final Justification:**

The rebuttal has addressed my concerns, so I raise my score.

**Key Questions For Authors:**

- For the results presented in Table 2, is my understanding correct that a ratio of 1.0 means that the obtained objective value is exactly identical to the ground truth?
- Still on the results presented in Table 2, is the reported time only consist of the time required to solve the resulting convex program for all approaches using Gurobi? Is the comparison for the total time end-to-end (including the training using LEMON for DFN) available?
- I understand that the number of parameters for all baselines are set to be identical, but is there a significant trade off to lowering the parameter count for the two baselines? Considering the MSE reported in Table 1 is very small (esp. for MDVSP), it will be very helpful to see at what point the speedup gained from lower parameter numbers will incur significant degradation in the MSE.

**Limitations:**

Yes

**Strengths And Weaknesses:**

#### Strength
- The paper is equipped with solid theoretical grounding and extensive analysis that illustrates the merits of DFN.
- The experimental design covers multiple settings, further supporting its primary claim of DFN's utility. Additionally, the results of those experiments shows favorable performance and efficiency.

#### Weaknesses
- It is very difficult to assess the position of the proposed DFN among existing bodies of work. This is mainly due to the absence of a dedicated section for existing/related works. While the introduction touches briefly on related topics such as common function approximators and the application of discrete functions, they are not sufficient to provide readers with a clear grasp on the topic discrete function approximators and what gap DFN is trying to fill among existing ones, if any.
- More notably, a dedicated explanation for the optimal flow cost problem is also absent both on the main text and the appendix. Considering that it is the main inspiration for DFN, even a short primer on the topic will immensely help readers understand the virtue of adapting it as a discrete function approximator.
- Considering the weaknesses mentioned above w.r.t. the presentation, the paper will benefit greatly from:
	- Allocating dedicated section that explain past works and recent advances of discrete function approximations and/or integer programming
	- Providing explanations on the optimal flow cost problem, more notably how to intuit that its general form is a good proxy for discrete function approximations
	- Moving more space-consuming proofs to the appendix to make room for additional discussions and experiments
- Another consequence of the weakness above is that it's hard to assess the significance of the presented result. Many of the works cited on the introduction is relatively old, it may be helpful to explain how DFN fits among more recent ML-guided approaches to MILP, for example [1].
- While the paper includes a solid theoretical analysis that provides a new perspective and application of the minimum-cost flow problem, the level of novelty is not so clear due to the lack of comparison to the original problem.

[1] Huang, W., Huang, T., Ferber, A. M., & Dilkina, B. (2024). Distributional MIPLIB: a multi-domain library for advancing ml-guided milp methods. _arXiv preprint arXiv:2406.06954_.

---

> ### Author Rebuttal · Authors · 2026-03-31
>
> **[W1]**
>
> - Our paper proposes a surrogate family for predict-then-optimize with integer downstream optimization. The introduction first motivates predict-then-optimize, then reviews relevant surrogate families for continuous variables, and finally explains the need for an analogous surrogate family for integer variables. To make our novelty claim more explicit, we will add the following sentence at the end of the fifth paragraph: "To the best of our knowledge, existing predict-then-optimize approaches over integer decisions typically rely on generic surrogate families, rather than surrogates specifically designed for downstream integer optimization."
> - As requested by the reviewer, we will also add an "Additional Related Literature" section; see [VBCU:W3]. This section will cover two adjacent strands of work that we initially omitted from the introduction in order to keep the paper focused on predict-then-optimize over integer decisions.
>
> **[W2]**
>
> Convex extendability alone is generally too weak to yield useful optimization guarantees, which is why one needs structured subfamilies that still allow efficient optimization. Minimum-cost-flow value functions are one such subfamily that remain algorithmically tractable while still rich enough to approximate complex data; please see [VBCU:W3].
>
> **[W3]**
>
> We will add the following "Additional Related Literature" section after the introduction:
>
> One related strand of literature is on function approximators for set-valued or binary inputs, such as Deep Sets (Zaheer et al., 2017), Deep Submodular Functions (Dolhansky and Bilmes, 2016), and Extended Deep Submodular Functions (Hosseini et al., 2024). While these are discrete-function approximators, they target set-valued or binary domains rather than general integer domains. These are also not designed to facilitate post-training optimization, and lead to complex generic downstream optimization formulations.
>
> Another related strand of literature is on surrogate-optimization models, including Gaussian-process and Kriging models (Jones, Schonlau, and Welch, 1998), as well as radial-basis-function models (Gutmann, 2001). These models are, however, more commonly used in sequential loops that alternate model fitting, optimization, and new data acquisition than in the predict-then-optimize setting considered here.
>
> Our paper is also related to the literature on discrete convex optimization. Convex extendability alone is generally too weak to yield interesting optimization guarantees, since minimizing a convex-extendible discrete function over integer points is NP-hard in general (Gritzmann and Klee, 1989). Accordingly, several more structured subclasses have been studied, including submodular set functions (Lovász, 1983), $L$/$L^\natural$-convex and $M$/$M^\natural$-convex functions (Murota, 2003), and integrally convex functions (Favati and Tardella, 1990). Minimum-cost-flow value functions form a particularly important subfamily of $M$-convex functions (Murota, 2003), combining algorithmic tractability with sufficient richness to approximate complex data.
>
> **[W4]**
>
> Our method is not an ML-guided MILP method, but a surrogate-construction method for predict-then-optimize over integer decisions. To the best of our knowledge, this is the first work that connects surrogate learning and discrete convexity in a way that achieves good-quality learning and more tractable downstream integer optimization.
>
> **[W5]**
>
> Minimum-cost-flow value functions are used here as the building block of a learnable surrogate family for discrete objectives, rather than as the original problem to be solved directly. The novelty is precisely to use this structure to obtain a surrogate that can be trained from data while yielding a more tractable downstream integer optimization problem than generic surrogate families.
>
> **[Q1]**
>
> Yes, your understanding is correct.
>
> **[Q2]**
>
> The reported time in Table 2 only includes the downstream optimization time after training. It is not a convex program, but an MILP. We focus on this quantity because the paper is in a predict-then-optimize setting, where the surrogate is trained offline once and then used for repeated downstream optimizations. For end-to-end wall-clock time, see [Pkx4:Exp3].
>
> **[Q3]**
>
> After rerunning the exact same quadratic experiment as in the paper and varying only the baseline size, we found that shrinking MLP or LSET to obtain downstream solve times comparable to DFN leads to a clear loss in predictive accuracy. For MLP, the points closest to the DFN solve-time region have test MSE $1.9\times$--$2.4\times$ larger than the original MLP. For LSET, the corresponding increase is $1.5\times$--$2.4\times$ relative to the original LSET.
>
> > **Overall:** We will add the text referenced in [VBCU:W1, W3] to the main text, the supplementary experiments [MDq9:Exp1, Exp4; Pkx4:Exp2, Exp3] to the appendix, and the corresponding code.

---

> > ### Author Rebuttal · Reviewer_VBCU · 2026-04-02
> >
> > Thank you for the feedback.
> > I have raised my scores accordingly.

---

> > > ### Author Response · Authors · 2026-04-04
> > >
> > > Thank you for the update. We are glad your concerns have been adequately addressed.

---

### Official Review · Reviewer_pkx4 · 2026-03-07

**Soundness:** 3
**Presentation:** 4
**Significance:** 3
**Originality:** 4
**Overall Recommendation:** 4
**Confidence:** 4

**Summary:**

The paper proposes Deep Flow Networks (DFNs), a class of models for learning discrete objective functions that remain tractable for downstream optimization. The key idea is to represent the predicted value as the optimal value of a minimum cost flow problem whose node imbalances depend linearly on the input. In this way, the model maps an input vector to the cost of routing flow through a network.

The paper makes three main contributions. First, it proves a universality result showing that this class of functions can approximate any discrete function that admits a convex extension to continuous space, on a finite set of integer points, and up to an affine scaling. Second, it analyzes optimization properties of the learned function. When the imbalance mapping is the identity, the function belongs to the class of M convex functions and can be minimized efficiently by steepest descent-type updates. For general mappings, the paper proves a generalized exchange property and shows that the complexity of local search depends on a parameter that measures the deviation from this regime. Third, the paper describes a training approach that embeds the minimum cost flow problem as a differentiable layer and learns its parameters using gradient-based methods. Experiments show that DFNs achieve good approximation accuracy and produce optimization problems that are easier to solve than those induced by standard neural network surrogates.

**Compliance With Llm Reviewing Policy:**

Affirmed.

**Key Questions For Authors:**

1.	The theory introduces a parameter that measures how far the model is from the easy optimization regime. Did the authors evaluate this quantity during training, and does it correlate with downstream optimization time?
2.	How sensitive is training to the initialization of the imbalance mapping matrix near the identity?
3.	Do the authors expect the observed optimization advantages to hold for richer feasible sets or more complex combinatorial constraints?

**Limitations:**

Yes

**Strengths And Weaknesses:**

A main strength of the paper is the clear and well-motivated connection between structured optimization models and learned surrogate objectives. The representation based on minimum cost flow value functions is novel and interesting. The universality theorem provides a strong conceptual justification for this representation.

Another strength is the optimization analysis. The paper carefully connects the model to known results from discrete convex analysis in the identity case and then extends these ideas to more general mappings through a generalized exchange property. This provides useful insight into when the learned objective remains tractable.

The implementation is also coherent. Embedding the flow problem as a differentiable layer allows the parameters of the optimization model to be learned directly from data, and the design choices are well aligned with the theoretical analysis.

The experiments clearly demonstrate the approximation and optimization advantages of the proposed model, though they are conducted mainly in controlled settings. Additional evaluation on more complex real-world decision problems could further strengthen the empirical case.

Another limitation is that the theoretical parameter that measures deviation from the easy optimization regime is not analyzed empirically. It would be useful to see how this quantity evolves during training and how strongly it correlates with optimization difficulty.

Overall, the paper presents a clear idea that combines representation theory, optimization structure, and differentiable training in a coherent way.

---

> ### Author Rebuttal · Authors · 2026-03-31
>
> **[W1]**
>
> We agree with the reviewer that it would be interesting to test the method on more complex industrial data as well. At the same time, our experiments were intentionally designed to reflect realistic settings while remaining controlled enough to isolate approximation quality and downstream optimization performance. Also, our evaluation on resource allocation and public multiple-depot vehicle scheduling (MDVSP) benchmarks is for practically relevant problems. We also refer the reviewer to [Pkx4:Exp3], where we test the method across a broader range of scales.
>
> **[W2]**
>
> Please see [MDq9:Exp1].
>
> **[Q1]**
>
> Please see [MDq9:Exp1].
>
> **[Q2]**
>
> This is an interesting question. We conducted an initialization-sensitivity study and found that, even under more randomized initializations, training remained stable and converged to good solutions; please see [Pkx4:Exp2].
>
> **[Q3]**
>
> We expect the optimization advantage to persist, because it comes from the fact that DFN leads to more structured constraint matrices.
>
> **[Exp2] Initialization sensitivity.**
>
> **Objective.** We study how sensitive learning is to the initialization used in the main text. In particular, we test whether DFN still trains reliably under more randomized initialization schemes for $(A,b)$, $c$, and $u$.
>
> **Setup.** We compare the default initialization against three more randomized initialization schemes: $(A,b)$ uniform over $\{-5,\ldots,5\}$, $c$ uniform over $[-5,5]$, and $u$ uniform over $\{1,\ldots,5\}$. We also include the parameter-matched MLP and LSET baselines for reference.
>
> | Initialization / Model | Test MSE |
> | --- | --- |
> | Default init | $0.0031 \pm 0.0005$ |
> | $(A,b)\sim U(\{-5,\ldots,5\})$ | $0.0019 \pm 0.0003$ |
> | $c\sim U([-5,5])$ | $0.0039 \pm 0.0001$ |
> | $u\sim U(\{1,\ldots,5\})$ | $0.0023 \pm 0.0001$ |
> | MLP | $0.0132 \pm 0.0014$ |
> | LSET | $0.0830 \pm 0.0047$ |
>
> **Discussion.** All DFN variants train successfully, and all remain clearly more accurate than MLP and LSET. Across the different DFN initialization schemes, final test errors remain in the same low range, indicating that the method is not brittle to a particular choice of initialization. The initialization used in the main text remains our preferred choice, since it is motivated not only by predictive performance but also by keeping the model closer to the easy optimization regime.
>
> **[Exp3] Additional scales of data and model size.**
>
> **Objective.** We study how training time, downstream optimization time, and test MSE scale as the data dimension and model size increase. In particular, we evaluate larger quadratic instances while keeping model sizes approximately matched across DFN, a variant of DFN with $A$ fixed at its near-identity initialization (DFN-fixedA), MLP, and LSET.
>
> **Setup.** We consider four scales: small $(d=8,K=1000,\approx 5$k params), medium $(d=16,K=2000,\approx 25$k), large $(d=24,K=4000,\approx 50$k), and xlarge $(d=32,K=6000,\approx 100$k). Downstream solve times use a one-hour time limit.
>
> *Training time (s):*
>
> | Model | Small | Medium | Large | XLarge |
> | --- | --- | --- | --- | --- |
> | DFN | $108.8 \pm 8.9$ | $597.9 \pm 28.0$ | $2161.7 \pm 182.4$ | $6443.1 \pm 957.0$ |
> | DFN-fixedA | $122.6 \pm 3.4$ | $1005.0 \pm 3.9$ | $3347.5 \pm 27.8$ | $9424.3 \pm 168.6$ |
> | MLP | $27.8 \pm 1.0$ | $66.5 \pm 2.1$ | $248.1 \pm 35.5$ | $734.1 \pm 69.4$ |
> | LSET | $32.0 \pm 1.2$ | $89.1 \pm 3.8$ | $264.7 \pm 33.9$ | $672.1 \pm 77.4$ |
>
> *Optimization time (s):*
>
> | Model | Small | Medium | Large | XLarge |
> | --- | --- | --- | --- | --- |
> | DFN | $0.068 \pm 0.011$ | $1.18 \pm 0.20$ | $15.74 \pm 1.57$ | $815.41 \pm 241.38$ |
> | DFN-fixedA | $0.050 \pm 0.008$ | $0.212 \pm 0.005$ | $0.570 \pm 0.118$ | $0.758 \pm 0.090$ |
> | MLP | $1.29 \pm 0.49$ | $55.01 \pm 35.42$ | $3563.43 \pm 36.87$ | $3600.48 \pm 0.04$ |
> | LSET | $1560.47 \pm 1034.97$ | $2656.78 \pm 943.83$ | $3600.60 \pm 0.23$ | $3165.12 \pm 435.39$ |
>
> *Test MSE:*
>
> | Model | Small | Medium | Large | XLarge |
> | --- | --- | --- | --- | --- |
> | DFN | $0.0108 \pm 0.0070$ | $0.0033 \pm 0.0005$ | $0.0040 \pm 0.0009$ | $0.0032 \pm 0.0002$ |
> | DFN-fixedA | $0.0563 \pm 0.0168$ | $0.0698 \pm 0.0015$ | $0.0734 \pm 0.0095$ | $0.0691 \pm 0.0054$ |
> | MLP | $0.0067 \pm 0.0010$ | $0.0127 \pm 0.0015$ | $0.0207 \pm 0.0009$ | $0.0164 \pm 0.0025$ |
> | LSET | $0.0366 \pm 0.0049$ | $0.0862 \pm 0.0081$ | $0.1417 \pm 0.0070$ | $0.1265 \pm 0.0143$ |
>
> **Discussion.** DFN is about one order of magnitude slower to train than MLP and LSET, which is the main computational cost of our approach. However, across scales, DFN retains strong approximation quality and a large downstream optimization advantage. The fixed-$A$ variant is even faster to optimize, at the cost of lower approximation quality.
>
> > **Overall:** We will add the text referenced in [VBCU:W1, W3] to the main text, the supplementary experiments [MDq9:Exp1, Exp4; Pkx4:Exp2, Exp3] to the appendix, and the corresponding code.

---

> > ### Author Rebuttal · Reviewer_pkx4 · 2026-04-03
> >
> > The additional experiments are helpful and move in the right direction toward addressing the concern about the role of the imbalance matrix A. In particular, Exp1 provides useful evidence that as A moves away from its identity-like initialization, downstream optimization becomes more difficult, which is consistent with the theoretical intuition. Similarly, Experiments 3 and 4 offer further insight into the practical importance of initializing A near the identity and suggest that this choice improves optimization behavior.
> >
> > Overall, these additions partially resolve my earlier questions by demonstrating a clear qualitative relationship between the structure of A and the optimization performance.
> >
> > That said, some aspects remain open and could be interesting directions for future work. In particular, it would be valuable to better understand the training dynamics of A, including how sensitive learning is to different initializations, whether A consistently drifts away from the identity during training, and how these effects relate to the theoretical complexity parameter. A more direct empirical characterization of this connection could further strengthen the theory–practice link.

---

> > > ### Author Response · Authors · 2026-04-04
> > >
> > > Thank you for your thoughtful follow-up! We are glad the added experiments helped clarify your concerns. One small clarification: Exp2 was intended to address initialization sensitivity directly, by comparing several different initializations of $A$ (as well as of $b$, $c$ and $u$) and showing that DFN still trains reliably across these variants. Exp1 was intended to address the training dynamics of $A$ and their connection to the theory, by tracking the evolution of $A$ across training checkpoints and showing that, in the studied setting, larger deviation from the identity-like initialization is accompanied by both higher downstream solve time and larger values of the complexity-related quantity, consistent with the theoretical intuition. We agree with the reviewer that studying whether similar dynamics are also present across other datasets and DFN architectures would be an interesting future direction building on our work.

---

### Official Review · Reviewer_MDq9 · 2026-03-13

**Soundness:** 3
**Presentation:** 4
**Significance:** 3
**Originality:** 3
**Overall Recommendation:** 5
**Confidence:** 3

**Summary:**

This paper introduces Deep Flow Networks (DFNs) as discrete function approximators param-
eterized by a minimum-cost flow instance $\Pi = (G, c, u, S, A, b)$. The authors show that the
proposed architecture serves as a universal approximator for discrete functions. Moreover, the
paper establishes that DFNs are suitable for solving certain optimization problems: in particular,
the time complexity of minimizing the induced DFN function $f (·; \Pi)$ is polynomial in quantities
related to the co-rank and pseudo-determinant of the matrix (A). Finally, the authors provide
empirical evaluations demonstrating the efficiency and accuracy of the proposed method across
several datasets, comparing its performance with existing approaches.

**Compliance With Llm Reviewing Policy:**

Affirmed.

**Final Justification:**

I appreciate the authors’ careful rebuttal, which fully addresses all of my comments. Accordingly, I have increased my score.

**Key Questions For Authors:**

- In many real-world applications, sampled data typically contain noise. How robust is the
proposed method when the target data are noisy? In particular, can the authors comment on
whether the DFN framework can still approximate the underlying function accurately and
perform optimization effectively in the presence of noise, either from a theoretical perspective
or through empirical evaluation?
- It would be helpful to understand how far the matrix A deviates from the identity matrix
I. Could the authors report a quantitative measure of this deviation in the appendix to
provide additional insight into the structure of A in the experiments?

**Minor comments**

- In the definition of DFNs, the matrix $A \in \mathbb{Z}^{p\times d}$ is introduced, but the meaning of d is not
explicitly defined at that point. It is later described as the dimension of the input. For clarity, it would be helpful to define d when the notation is first introduced.
- The notation $[p]$ is used but not explicitly defined in the notation section.
- In equation (1a), the vector e appears; it may be clearer to define it explicitly as ei to avoid
ambiguity.

**Limitations:**

yes

**Strengths And Weaknesses:**

**Strengths**
- The paper is generally well organized and clearly written. The exposition is structured
in a logical manner, with definitions, lemmas, and theorems introduced in a coherent or-
der and supported by accompanying explanations. The connections between sections are
clearly established, allowing the reader to follow the development of the proposed framework
step by step. Overall, the manuscript builds the necessary theoretical foundations before
progressively developing the main results.
- To the best of my understanding, the theoretical claims appear to be technically correct and
supported by appropriate analysis. The assumptions underlying the results seem reasonable
within the scope of the paper. In addition, the empirical evaluation appears to be conducted
in a logical and methodologically appropriate manner, and the experiments are designed to
illustrate the behavior of the proposed approach.
- The paper introduces a generalized framework based on minimum-cost flow value functions and shows theoretically that the proposed model can approximate a class of discrete functions efficiently. This perspective provides an interesting connection between discrete
optimization and function approximation. While the core components build on existing ideas from network flow theory and deep learning, their combination into the proposed Deep Flow Network framework appears novel and is supported by both theoretical analysis
and empirical evaluation.

**Weaknesses**

- Although the proposed method is technically well motivated and contributes within its specific context, the overall impact appears somewhat limited. The contribution mainly applies to a specialized setting involving minimum-cost flow value functions, which may restrict its broader applicability within the machine learning community. As a result, although the work provides interesting insights for this particular domain, its potential influence on a wider range of ML problems or applications seems moderate.
- A substantial portion of the manuscript is devoted to establishing theoretical results showing
that Algorithm 1 finds a global minimizer of $f (·; \Pi)$ and terminates in polynomial time.
However, in the empirical evaluation, the optimization appears to be performed using integer
programming rather than Algorithm 1, and no complexity analysis is provided for this
implementation beyond the simulation results. This raises some uncertainty regarding the
practical role of Algorithm 1 and how closely the empirical procedure reflects the theoretical
part analyzed in the paper.
- In the simulation setup, the datasets (specifically yt) appear to be generated from a mini mization problem similar to the structure assumed by the proposed method. This raises a potential concern about the fairness of the comparison with more general function approximators such as MLPs, which are not explicitly designed for this structured optimization
setting.

---

> ### Author Rebuttal · Authors · 2026-03-31
>
> **[W1]**
>
> - We would like to clarify that DFN targets the broader and richer class of convex-extendible discrete functions, not only data generated by min-cost-flow instances; the quadratic dataset in our experiments is a direct example.
> - The min-cost-flow class is only used to define a learnable surrogate for discrete objectives, and was chosen because it is both rich and easy to optimize over integers; see also [VBCU:W2].
>
> **[W2]**
>
> Please see [Zxzm:W2].
>
> **[W3]**
>
> - We note that the quadratic dataset in our experiments is not generated from a min-cost-flow value function.
> - More broadly, our results show that DFN is a strong surrogate choice for convex-generated data. We also included LSET precisely because it has a convexity inductive bias, so the comparison is not only against a generic MLP but also against another convex-data approximator.
>
> **[Q1]**
>
> This is an interesting question. We conducted a dedicated noise-robustness experiment; please see [MDq9:Exp4].
>
> **[Q2]**
>
> This is also a very interesting question. We conducted a dedicated experiment on the evolution of $A$; please see [MDq9:Exp1].
>
> **Supplementary Experiments**
>
> Unless noted otherwise, all experiments use the convex quadratic dataset setting from the main text, the same 1000-epoch training budget, and approximately parameter-matched baselines. All reported quantities are mean $\pm$ standard error over $3$ seeds.
>
> **[Exp1] Evolution of $A$ and downstream optimization time.**
>
> **Objective.** We study how downstream optimization difficulty changes as the learned matrix $A$ moves away from its identity-like initialization $A_0$, which is designed to keep DFN near the easy optimization regime.
>
> **Setup.** We train the DFN used for the convex quadratic dataset setting in the main text and save checkpoints every 100 epochs. To isolate the effect of $A$, we take the final trained model, keep all non-$A$ parameters fixed, and replace only $A$ by the checkpoint value before solving the downstream optimization problem. We report both $\|A-A_0\|_1$ and $\log \operatorname{pdet}(A)$.
>
> | Epoch | $\|A-A_0\|_1$ | $\log \operatorname{pdet}(A)$ | Solve time (s) |
> | --- | --- | --- | --- |
> | 100 | $1.67 \pm 0.33$ | $11.34 \pm 0.09$ | $0.113 \pm 0.008$ |
> | 200 | $110.67 \pm 34.89$ | $15.04 \pm 1.26$ | $0.177 \pm 0.011$ |
> | 300 | $111.00 \pm 35.04$ | $15.04 \pm 1.26$ | $0.201 \pm 0.015$ |
> | 400 | $186.33 \pm 16.76$ | $20.23 \pm 1.04$ | $0.218 \pm 0.020$ |
> | 500 | $720.67 \pm 74.06$ | $33.35 \pm 1.55$ | $0.305 \pm 0.037$ |
> | 600 | $1278.67 \pm 54.18$ | $41.30 \pm 0.96$ | $0.468 \pm 0.063$ |
> | 700 | $1694.67 \pm 33.94$ | $45.37 \pm 0.77$ | $0.596 \pm 0.122$ |
> | 800 | $2034.00 \pm 38.35$ | $48.04 \pm 0.73$ | $1.350 \pm 0.201$ |
> | 900 | $2291.33 \pm 21.93$ | $49.88 \pm 0.52$ | $1.176 \pm 0.176$ |
> | 1000 | $2506.33 \pm 29.08$ | $51.35 \pm 0.53$ | $3.235 \pm 1.055$ |
>
> **Discussion.** Both measures of deviation increase over training, and the downstream solve time rises with them. Solve time is strongly correlated with both $\|A-A_0\|_1$ and $\log \operatorname{pdet}(A)$, which is consistent with the theory that moving farther away from the identity-like regime makes optimization harder.
>
> **[Exp4] Noise robustness.**
>
> **Objective.** We study whether DFN remains reliable when the labels are noisy. In particular, we measure how the approximation quality degrades as observation noise is added to the target values.
>
> **Setup.** We use the same quadratic dataset as in the main text and the same models. For each seed, we compute the clean-label standard deviation and inject Gaussian noise with standard deviation equal to $\{0,0.05,0.10,0.20\}$ times that quantity. We then retrain DFN, MLP, and LSET under the same training budget as in the main text.
>
> | Noise ratio | DFN MSE | MLP MSE | LSET MSE |
> | --- | --- | --- | --- |
> | 0.00 | $0.0031 \pm 0.0005$ | $0.0132 \pm 0.0014$ | $0.0830 \pm 0.0047$ |
> | 0.05 | $0.0070 \pm 0.0006$ | $0.0167 \pm 0.0002$ | $0.0867 \pm 0.0046$ |
> | 0.10 | $0.0176 \pm 0.0002$ | $0.0294 \pm 0.0004$ | $0.0962 \pm 0.0054$ |
> | 0.20 | $0.0541 \pm 0.0009$ | $0.0749 \pm 0.0019$ | $0.1225 \pm 0.0078$ |
>
> **Discussion.** All models degrade smoothly as the noise level increases, with no failed DFN runs. DFN remains the most accurate model at every noise level, and the gap over MLP and LSET persists even at the highest noise level we tested. This suggests that our training procedure is stable and that the architecture is robust to moderate label noise.
>
> > **Overall:** We will add the text referenced in [VBCU:W1, W3] to the main text, the supplementary experiments [MDq9:Exp1, Exp4; Pkx4:Exp2, Exp3] to the appendix, and the corresponding code.

---

> > ### Author Rebuttal · Reviewer_MDq9 · 2026-04-04
> >
> > I appreciate the authors’ careful rebuttal, which fully addresses all of my comments. Accordingly, I have increased my score.

---

> > > ### Author Response · Authors · 2026-04-04
> > >
> > > Thank you! We appreciate your update and your positive assessment.

---

### Official Review · Reviewer_Zxzm · 2026-03-13

**Soundness:** 3
**Presentation:** 3
**Significance:** 3
**Originality:** 4
**Overall Recommendation:** 5
**Confidence:** 3

**Summary:**

The paper introduces Deep Flow Networks (DFNs), a novel architecture designed to approximate convex-extendible discrete functions. A DFN maps integer inputs to the optimal cost of a parameterized minimum-cost flow problem, where the inputs determine the net outflows of specific nodes via an affine transformation. The authors provide theoretical results demonstrating that DFNs are universal approximators for this class of functions. Furthermore, they characterize the complexity of optimizing a trained DFN over integer domains based on the learned matrix's deviation from the tractable M-convex regime. Empirically, the model is trained using Straight-Through Estimation (STE) for discrete parameters and evaluated on three small-scale tasks (quadratic, resource allocation, and multi-depot vehicle scheduling). The results show improved test MSE and faster downstream integer optimization times compared to standard multi-layer perceptron (MLP) and LogSumExp (LSET) baselines.

**Compliance With Llm Reviewing Policy:**

Affirmed.

**Final Justification:**

My concerns have been adequately addressed - the authors answered my questions with new experimental data.

**Key Questions For Authors:**

1. How does the training time of DFNs compare to the baselines (MLP, LSET) in terms of actual wall-clock time?
2. Can the authors provide empirical evidence or a discussion on the stability of the STE approach for this architecture? Were there issues with dead nodes, zero gradients, or diverging parameters during training?
3. How does the downstream integer program's solve time scale as the number of layers or layer widths in the DFN increases? At what network size does the exact solver begin to struggle?

**Limitations:**

The authors acknowledge the training time bottleneck and mention future work on batch solving and approximate min-cost flow algorithms. However, the limitation of relying on exact commercial solvers (e.g., Gurobi) for the downstream optimization of the DFN itself is not thoroughly discussed. The paper would benefit from a clearer acknowledgment of the trade-off between the model's expressivity (size) and the practical limits of solving the resulting IP at inference time.

**Strengths And Weaknesses:**

## Strengths
- The approach is novel and theoretically well-grounded. The connection between network flow value functions, M-convexity, and surrogate modeling for discrete optimization is elegant.
- The theoretical bound on optimization complexity (via the generalized exchange property and the $\Delta(A)$ parameter) provides rigorous insights into the tractability of the proposed approximators.
- The method demonstrates improved downstream optimization efficiency on the evaluated benchmarks compared to standard ReLU MLPs and LSET models, highlighting its potential utility in predict-then-optimize pipelines.

## Weaknesses
- The forward pass requires solving an exact minimum-cost flow problem for each sample. The experiments utilize a very small batch size of 8 and a relatively compact network (three layers, ~26k parameters) trained for only 1000 epochs. It is unclear how this training procedure can scale to larger datasets or higher-dimensional inputs, as exact solvers will likely become a severe computational bottleneck.
- While theoretically analyzed via a local descent algorithm, the practical downstream optimization in the experiments relies on passing an equivalent IP to Gurobi. As the DFN depth or width grows, the resulting IP will grow accordingly, which may negate the tractability benefits compared to simpler surrogate models.
- The empirical evaluation is restricted to three relatively low-dimensional (e.g., $d=16$) synthetic or semi-synthetic datasets. The baselines are limited to basic MLPs and LSET. Evaluating on established, larger-scale predict-then-optimize benchmarks would provide a more convincing demonstration of the architecture's practical value.
- Relying on Straight-Through Estimation (STE) to learn the integer components of the affine transformation and the capacities can introduce training instability, particularly because gradients are propagated through an exact IP solver.

---

> ### Author Rebuttal · Authors · 2026-03-31
>
> **[W1]**
>
> - We recognize that the main strength of our model is not training speed; rather, it is the combination of good approximation quality with much easier downstream integer optimization than generic surrogates such as MLP/LSET.
> - To further clarify how the model behaves across scales, we conducted additional scaling experiments; please see [Pkx4:Exp3]. In brief, the scaling experiments show that DFN is about $9\times$ slower than MLP to train across the tested scales, but it retains clear advantages in downstream optimization and test MSE; we also expect the training overhead to decrease with further engineering.
> - Optimization layers in the forward pass are common in the literature, for example in OptNet (Amos and Kolter, 2017) and differentiable convex optimization layers such as cvxpylayers (Agrawal et al., 2019). Notably, in our setting the layer is a highly structured min-cost-flow problem. This allows us to use the specialized LEMON solver instead of a generic LP solver, which we observed to be significantly faster than generic solvers such as the LP solver of Gurobi on CPU and the LP solver of NVIDIA cuOpt on an H100 GPU.
> - Finally, in ongoing work we are exploring extensions that could push beyond the current scale regime. In particular, a two-layer DFN is effectively an assignment problem, for which Sinkhorn-type methods can be extremely fast since they are largely matrix-multiplication based. This is however beyond the scope of the present paper, which is meant to introduce the general architecture, theory, and training method for medium-scale problems.
>
> **[W2]**
>
> - The theoretical local-descent algorithm and the practical Gurobi solver play different roles:
> - The local-descent result gives a complexity guarantee: when $\Delta(A)$ is small, minimizing a DFN admits a pseudo-polynomial optimization guarantee.
> - In practice, we solve the equivalent DFN IP with Gurobi and observe that it is significantly faster than the corresponding MLP/LSET formulations. As we mention in the paper, the reason is that the DFN constraint matrix has the form $[N,A]$, where $N$ is a network matrix and therefore totally unimodular, while $A$ acts only as a small correction.
> - This is also supported by [MDq9:Exp1], where we observe a clear empirical correlation between downstream solve time and the theoretical complexity parameter $\Delta(A)$.
>
> **[W3]**
>
> - For additional results across scales, see [Pkx4:Exp3].
> - For baselines, we chose MLP because it is the most widely used neural baseline in predict-then-optimize. More specifically, it is typically used by replacing part of an optimization model with a learned neural surrogate and then solving the resulting formulation, often as a MILP. We expect DFN to be used in the same way, but to yield a more structured downstream MILP. We chose LSET because it is a standard model for learning continuous convex data that still admits a generic solver formulation.
>
> **[W4]**
>
> - We first clarify that, during training, the forward pass solves a minimum-cost-flow problem. When supplies and capacities are integral, the corresponding LP already has integral optimal solutions, so the backward pass is through a structured LP rather than an IP solver.
> - Empirically, we did not observe failed or divergent DFN training runs. In fact, we also conducted an initialization-sensitivity study and found that, even under different and more randomized initializations, training remained stable and converged to good solutions; see [Pkx4:Exp2].
>
> **[Q1]**
>
> We report this in [Pkx4:Exp3]. In brief, DFN training is about $9\times$ slower than MLP and LSET across the tested scales.
>
> **[Q2]**
>
> No, we did not observe dead nodes, zero gradients, or diverging parameters during training; please see [Zxzm:W4].
>
> **[Q3]**
>
> We conducted experiments over a range of problem sizes and observed that downstream IP time grows with scale, but DFN remains consistently faster than MLP and LSET. When $A$ is fixed to its near-identity initialization, optimization is even significantly faster across all scales. Please see [Pkx4:Exp3].
>
> **[L1]**
>
> While our approach ultimately embeds the learned surrogate within an IP, its main advantage is precisely that the resulting IPs are much faster to solve than those induced by existing benchmark surrogates, as shown in our experiments. This suggests that DFN can help speed up downstream IP formulations arising in predict-then-optimize applications with discrete variables, for example in transportation, inventory, and logistics.
>
> > **Overall:** We will add the text referenced in [VBCU:W1, W3] to the main text, the supplementary experiments [MDq9:Exp1, Exp4; Pkx4:Exp2, Exp3] to the appendix, and the corresponding code.

---

> > ### Author Rebuttal · Reviewer_Zxzm · 2026-04-04
> >
> > Thank you for the detailed rebuttal. I have updated the score accordingly.

---

> > > ### Author Response · Authors · 2026-04-04
> > >
> > > Thank you for the update! We are glad the rebuttal addressed your concerns.

---

### Decision · Program_Chairs · 2026-04-30

**Decision:**

Accept (spotlight)

**Comment:**

All reviewers recommend acceptance, and I accept their recommendation. The paper introduces a novel and elegant contribution, supported by strong theoretical results and coherent empirical evaluation. Reviewers commended the originality of the framework, the clarity of the theoretical development, and the practical design choices. The main concerns, namely, limited experimental scale, narrow baselines, and presentation gaps around related work, were substantively addressed in the rebuttal. I encourage the authors to incorporate all promised revisions into the final version.